# MassSpecGym: A benchmark for the discovery and identification of molecules

**Roman Bushuiev**[1,2], **Anton Bushuiev**[2], **Niek F. de Jonge**[3], **Adamo Young**[4],
**Fleming Kretschmer**[5], **Raman Samusevich**[1,2], **Janne Heirman**[6], **Fei Wang**[7,8],
**Luke Zhang**[9], **Kai Dührkop**[5], **Marcus Ludwig**[10], **Nils A. Haupt**[5], **Apurva Kalia**[11],
**Corinna Brungs**[1], **Robin Schmid**[1], **Russell Greiner**[7,8], **Bo Wang**[4], **David S. Wishart**[7,12],
**Li-Ping Liu**[11], **Juho Rousu**[13], **Wout Bittremieux**[6], **Hannes Rost**[9], **Tytus D. Mak**[14],
**Soha Hassoun**[11,15], **Florian Huber**[16], **Justin J.J. van der Hooft**[3,17], **Michael A. Stravs**[18],
**Sebastian Böcker**[5], **Josef Sivic**[2], **Tomáš Pluskal**[1]

[1]Institute of Organic Chemistry and Biochemistry of the Czech Academy of Sciences,
[2]Czech Institute of Informatics, Robotics and Cybernetics, Czech Technical University,
[3]Bioinformatics Group, Wageningen University & Research, [4]Department of Computer
Science, University of Toronto, [5]Chair for Bioinformatics, Institute for Computer Science,
Friedrich Schiller University Jena, [6]Department of Computer Science, University of Antwerp,
[7]Department of computing science, University of Alberta, [8]Alberta Machine Intelligence Institute,
[9]Department of Molecular Genetics, University of Toronto, [10]Bright Giant GmbH, [11]Department of
Computer Science, Tufts University, [12]Department of Biological Sciences, University of Alberta,
[13]Department of Computer Science, Aalto University, [14]Mass Spectrometry Data Center,
National Institute of Standards and Technology, [15]Department of Chemical and Biological
Engineering, Tufts University, [16]Centre for Digitalisation and Digitality, University of Applied
Sciences Düsseldorf, [17]Department of Biochemistry, University of Johannesburg,
[18]Eawag: Swiss Federal Institute of Aquatic Science and Technology

## Abstract

The discovery and identification of molecules in biological and environmental
samples is crucial for advancing biomedical and chemical sciences. Tandem mass
spectrometry (MS/MS) is the leading technique for high-throughput elucidation
of molecular structures. However, decoding a molecular structure from its mass
spectrum is exceptionally challenging, even when performed by human experts. As
a result, the vast majority of acquired MS/MS spectra remain uninterpreted, thereby
limiting our understanding of the underlying (bio)chemical processes. Despite
decades of progress in machine learning applications for predicting molecular
structures from MS/MS spectra, the development of new methods is severely
hindered by the lack of standard datasets and evaluation protocols. To address
this problem, we propose MassSpecGym – the first comprehensive benchmark
for the discovery and identification of molecules from MS/MS data. Our bench-
mark comprises the largest publicly available collection of high-quality labeled
MS/MS spectra and defines three MS/MS annotation challenges: *de novo* molecu-
lar structure generation, molecule retrieval, and spectrum simulation. It includes
new evaluation metrics and a generalization-demanding data split, therefore stan-
dardizing the MS/MS annotation tasks and rendering the problem accessible to
the broad machine learning community. MassSpecGym is publicly available at
`https://github.com/pluskal-lab/MassSpecGym`.

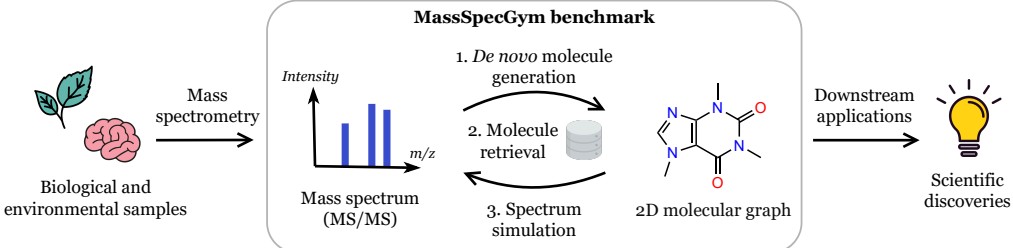

Figure 1: **MassSpecGym provides three challenges for benchmarking the discovery and identification of new molecules from MS/MS spectra**. The provided challenges abstract the process of scientific discovery from biological and environmental samples into well-defined machine learning problems.

# 1   Introduction

The discovery and identification of small molecules profoundly influence numerous scientific fields, including organic chemistry [1], molecular biology [2], drug development [3], disease diagnosis [4], environmental analysis [5], and space exploration [6]. Despite significant progress, it is estimated that only a small fraction of molecules across the kingdoms of life have been discovered [7]. Tandem mass spectrometry (MS/MS) is the most widely used technique for elucidating molecular structures from biological and environmental samples, supporting a wide range of applications in biotechnology and medicine [8]. In drug development, MS/MS is crucial for identifying novel bioactive compounds [9], such as those targeting cancer and infectious diseases [7]. MS/MS also plays a key role in clinical settings for determining appropriate drug dosages and assessing potential side effects [10]. In environmental analysis, it enables the detection of pollutants at trace levels, which is vital for monitoring and preserving environmental health [11]. Moreover, MS/MS addresses various challenges in structural biology, including the discovery of ligands that bind to target proteins [12] and the elucidation of metabolic pathways [13].

When analyzing a sample, a mass spectrometer typically generates thousands of tandem mass spectra, each characterizing a specific molecule present in the sample. While the annotation of mass spectra with molecular structures is inherent to mass spectrometry, it remains a significant challenge. From typical samples of interest, typically less than 10% of MS/MS spectra are annotated using state-of-the-art methods [14, 15]. As a result, the natural chemical space remains largely unexplored, thereby hindering scientific advancements.

To generate an MS/MS spectrum, a mass spectrometer follows an intricate multi-step procedure. First, the instrument ionizes the molecule using methods such as electrospray ionization (ESI). During this process, the molecule gains additional atoms, known as the ionization adduct. Subsequently, the ionized molecule (often referred to as precursor ion) is fragmented using collision-induced dissociation (CID), higher energy collisional dissociation (HCD), or other fragmentation method [16]. Finally, for each individual fragment ion, the instrument records its (i) mass-to-charge ratio (m/z value; the charge is typically equal to one for small molecules) and (ii) its corresponding abundance (signal intensity). The collection of these two-dimensional data points, characterizing the molecule as a distribution of fragment masses, is referred to as a tandem mass spectrum, MS/MS spectrum, or $MS^2$ spectrum.

The most notable progress in MS/MS annotation has been achieved by machine learning methods augmented with combinatorial optimization and domain expertise [17, 18]. However, these methods have not seen significant improvements in recent years due to their lack of scalability and small return of increased human knowledge. In contrast, recent years have witnessed numerous purely data-driven deep learning models performing competitively or even surpassing the classic approaches [19, 20, 21, 22, 23, 24, 25, 26, 27]. Nevertheless, the development of this new generation of modern machine learning methods for MS/MS spectrum annotation is currently hindered by multiple factors. These factors include the heterogeneity of data acquired under different mass spectrometry settings, the scarcity of high-quality annotated spectra, variations in data pre-processing techniques, inconsistencies in data splitting methods resulting in data leakage, differences in approaches to MS/MS annotation, varying evaluation metrics, and the proprietary nature of many datasets. As a result, developing a machine learning algorithm for mass spectrum annotation currently necessitates

mass spectrometry domain expertise, rigorous data preparation, and the reevaluation of existing methods for benchmarking purposes.

At the same time, dataset collection and benchmarking efforts have been one of the key drivers responsible for breakthrough progress in machine-learning-driven fields, for example: ImageNet [28], SQuAD [29], Gym [30], ProteinGym [31, 32], and OGB [33]. Inspired by these efforts, we propose MassSpecGym – a new public dataset of MS/MS spectra and a unified benchmarking protocol for MS/MS spectrum annotation (Figure 1). Our dataset provides a standardized collection of 231 thousand high-quality mass spectra representing 29 thousand unique molecular structures, making it the largest publicly available dataset. 10 thousand molecules (33%) present in the dataset are derived from our newly measured in-house data (i.e., MSnLib library presented in [34]). Additionally, we provide a curated selection of large unlabeled datasets of mass spectra and molecules allowing for the combination of supervised and unsupervised methods [20, 35]. Importantly, we develop a new splitting procedure based on the edit distance of molecular structures and divide our dataset into non-leaking train-validation-test folds. The MassSpecGym benchmark defines three MS/MS annotation challenges: *de novo* molecular structure generation, molecule retrieval, and spectrum simulation. We make each of the challenges easily accessible to the broad machine learning community by providing MassSpecGym through a user-friendly interface leveraging PyTorch Lightning and Hugging Face platforms[1]. Users can build new models on top of the prepared components and submit their results to the *Papers With Code* leaderboard. We anticipate that our unified benchmark will have a significant impact on the community by enabling reproducible research and accelerating the development of new MS/MS spectrum annotation methods.

## 2 Related work

**Labeled MS/MS data.** The creation of spectral libraries is driven by the desire to facilitate the annotation of a measured query spectrum [39, 40]. A spectral library catalogues a molecule and one or more of its spectra that are measured under different mass spectrometry instrument conditions. There are in-house private spectral libraries, commercial libraries, and openly accessible crowd-sourced libraries. MassBank [38], MassBank of North America (MoNA) [37] and GNPS [36] are the three largest crowd-sourced libraries comprising tens of thousands of molecules in total. The National Institute of Standards and Technology (NIST) provides a variety of for-purchase spectral libraries comprising up to 52 thousand compounds. However, NIST libraries are not available for

Table 1: **MassSpecGym is the largest publicly available dataset of high-quality labeled MS/MS spectra.** Our quality assessment workflow eliminates noisy or corrupted spectra and ensures reliable molecular labels and metadata (Section 3.3). The "Split" column highlights that, unlike other large-scale datasets, MassSpecGym provides a pre-defined data split.

| Dataset | Spectra | High-quality spectra | Molecules | Split |
|---|---|---|---|---|
| GNPS [36] | **322K** | 104K | 16K | ✗ |
| MoNA [37] | 98K | 62K | 10K | ✗ |
| MassBank [38] | 62K | 58K | 4K | ✗ |
| MIST CANOPUS [19] | 11K | $\leq$ 11K | $\leq$ 9K | ✓ |
| MassSpecGym (ours) | 231K | **231K** | **29K** | ✓ |

machine learning due to licensing restrictions. A similar situation exists with mzCloud [41], which provides MS/MS spectra for 32 thousand compounds but cannot be downloaded and used outside its native web interface.

The availability of spectral libraries has provided labeled datasets for supervised machine learning, but there are many challenges. These libraries are relatively small, covering only thousands to tens of thousands of molecules. Consequently, many annotation tools combine data from various libraries, including proprietary sources, limiting reproducibility and introducing biases. Public crowd-sourced datasets often contain low-quality, noisy mass spectra or invalid metadata, necessitating custom pre-processing and filtering techniques. While these techniques aim to improve dataset quality, they often limit the applicability and reproducibility of the corresponding machine learning methods. Additionally, the heterogeneity and non-standardization of mass spectrometry instruments and parameters challenge effective learning from spectral libraries. Our MassSpecGym dataset offers the first carefully curated and standardized collection of MS/MS spectra, maintaining high quality and surpassing existing datasets in size (Table 1).

---

[1]`https://github.com/pluskal-lab/MassSpecGym`

**Train-validation-test splitting of MS/MS data.** Most of the previous studies split labeled MS/MS data such that molecules with identical planar structures do not appear in different training, validation, and test sets [17, 35, 24, 19, 21]. This is achieved by using distinct 2D InChiKey hash descriptions of molecules for each data fold. However, this method can be compromised by minor structural modifications often found in spectral libraries as a result of, for example, click chemistry [42]. Our MassSpecGym benchmark has undergone extensive vetting in terms of data splitting. In this work, to prevent data leakage and to accurately assess model generalization to novel molecules, we develop a data splitting strategy that guarantees that there are no leaks with the chemical bond edit distance (i.e., MCES distance [43]) less than 10 (Figure 2).

**Benchmarking MS/MS annotation.** Currently, there are no comprehensive and standardized datasets available for the development and evaluation of models predicting spectra or molecular structures. One recently utilized dataset for benchmarking is MIST CANOPUS [19, 35], which was curated to ensure an even distribution of chemical classes [44]. However, this dataset is relatively small, comprising only 9 thousand molecules and 11 thousand spectra, and employs a data split based on 2D InChIKey, a method resulting in data leakage (Figure 2).

The Critical Assessment of Small Molecule Identification (CASMI) series [45] is another example of a recent benchmarking initiative. However, the CASMI challenge is held only once every two years at best, limiting opportunities for continuous evaluation and benchmarking. Additionally, the CASMI datasets are relatively small, comprising several hundred spectra representing challenging test cases. Participation in the challenge also demands significant mass spectrometry domain expertise for preprocessing the data into the format suitable for machine learning. In contrast, our proposed MassSpecGym is based on the new largest publicly available dataset (Table 1) and is designed to be machine learning-ready, thereby addressing the limitations inherent in the MIST CANOPUS and CASMI benchmarks.

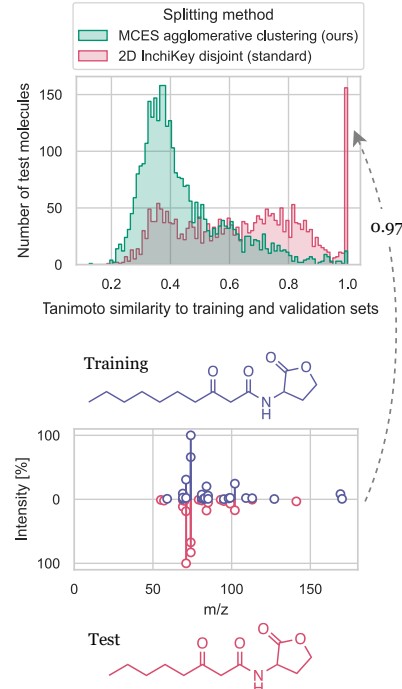

Figure 2: **Our MCES-based data split resolves the data-leakage issue abundant in prior work**. The standard approach separates molecules with identical planar structures (2D InChIKeys) into different folds, disregarding minor molecular modifications. This leads to near-duplicate test molecules (with Tanimoto similarity > 0.85) being leaked from the training data (red), as shown in the example below. In contrast, our approach maximizes molecular edit distance (MCES) between training and test sets, ensuring distinct data folds (green).

## 3 MassSpecGym benchmark

This section describes the construction of the MassSpecGym benchmark. First, we define three challenges of mass spectrum annotation along with the corresponding evaluation metrics (Section 3.2). Then, we describe the collection and processing of the underlying dataset of mass spectra and analyze its composition (Section 3.3). Finally, we outline our procedure for splitting the dataset into train-validation-test folds and demonstrate its generalization-demanding nature (Section 3.4). Please see the details on the construction of MassSpecGym in Supplementary Information.

### 3.1 Motivation for the challenges

***De novo* molecule generation.** The first challenge is the *de novo* prediction of a molecular graph from an MS/MS mass spectrum. This challenge can be compared to the goal of AlphaFold [46] but instead of predicting protein structures from their sequences, the task here is to predict small molecule structures from their MS/MS spectra. As such, this task represents a grand challenge in computational mass spectrometry, given its potential to drive the discovery of novel natural products, drug metabolites, environmental transformation products, and other crucial molecules [14]. A model

that can accurately predict molecular structures from MS/MS spectra could significantly advance our understanding of biology by enabling the annotation of metabolomes of uncharacterized organisms [7].

**Molecule retrieval.** The second challenge focuses on retrieving a molecular graph from a molecular database given a mass spectrum, rather than generating a completely new molecule. This scenario is common in practice when determining if a sample contains specific compounds, such as pesticides, environmental pollutants, or other known substances [11]. This approach is also relevant in drug design, particularly in affinity selection–mass spectrometry, where protein binders are identified from combinatorial libraries of ligands [12].

**Spectrum simulation.** The third challenge, called spectrum simulation, is the inverse problem of predicting a mass spectrum from a molecular graph. This task has two main motivations. First, it enhances the understanding of MS/MS fragmentation mechanisms in organic chemistry, leading to more precise predictions of how molecules will behave under various conditions. This insight can aid in the design of novel compounds and the optimization of synthetic pathways [47]. Second, it allows for pseudolabeling, expanding training datasets for machine learning models by generating synthetic spectra, which can improve model performance when experimental data is limited [48].

## 3.2 Definition of the challenges

*De novo* **molecule generation.** The task of *de novo* molecular generation is to generate a molecular structure from a mass spectrum. Formally, the input is a mass spectrum $X \subset \mathbb{R}_+ \times (0, 1]$, consisting of a set of two-dimensional points (referred to as signals or peaks) representing m/z (mass-to-charge) values and their corresponding intensities, which are normalized by dividing each by the maximum intensity. Intuitively, these points describe the abundance of molecular fragments with different masses. The goal is to generate a molecular graph $\hat{G} = (V, E)$, where vertices $V \in \mathbb{V}^N$, $|\mathbb{V}| = 118$ is a set of $N$ atoms from the vocabulary of 118 chemical elements (or, for example, 10 most common ones [24]) characterized by the periodic table, and $E \in \mathbb{E}^M$, $|\mathbb{E}| = 4$ is a set of $M$ edges from the vocabulary of 4 chemical bonds between atoms: single, double, triple, and aromatic [33]. Note that we do not model the 3D coordinates of chemical graphs, as the information in MS/MS spectra is typically insufficient for predicting exact molecular conformations [49].

Given the complexity of *de novo* generation, we propose an additional, simpler, challenge where chemical formulae are provided as input, meaning the set of vertices $V$ is known. In practice, chemical formulae can be derived with high accuracy by utilizing MS[1] mass spectra, an orthogonal data source to MS/MS [50, 51]. Since working with MS[1] data is typically based on combinatorial optimization rather than machine learning [52], our benchmark directly provides chemical formulae instead of MS[1] spectra, imitating the output of the MS[1] spectra processing pipelines. However, we present this scenario as a bonus challenge because chemical formula prediction remains a partially unsolved problem. For example, elements such as fluorine, which have only one stable isotope, cannot be derived from MS[1] data alone and still pose challenges with MS/MS data [20].

While each mass spectrum is a measurement on a specific compound, the spectrum may not contain all the necessary information to fully reconstruct the molecular structure as the spectrum is a partial view of the measured compound. Therefore, our approach acknowledges this complexity and permits multiple plausible molecular structures corresponding to a given spectrum. To this end, we formulate the problem as predicting a set of $k$ graphs $\hat{\mathcal{G}}_k = \{\hat{G}_1, \ldots, \hat{G}_k\}$ rather than a single solution $\hat{G}$. These graphs can be sampled randomly from a model or selected as the top-$k$ predictions from a larger set, if a scoring function is available. This approach reflects the inherent uncertainty and challenges of accurately predicting the correct molecular graph from spectral data.

We evaluate the correspondence between the generated molecular graphs $\hat{\mathcal{G}}_k$ and the ground-truth graph $G$ using three metrics. Ideally, the set of predictions includes the ground-truth graph, which we assess by measuring

$$\text{Top-}k \text{ accuracy: } \mathbb{1}\{G \in \hat{\mathcal{G}}_k\}, \tag{1}$$

averaged across all test examples. In the equation, $\mathbb{1}$ is the indicator function returning 1 if the condition is true and 0 otherwise. The top-$k$ accuracy varies between 0 and 1, where 0 corresponds to none of the test samples having the ground truth graph among the top-$k$ prediction and 1 corresponds

to all test samples having the ground truth graph among the top-$k$ predictions. Given the difficulty of predicting the exact graph, we also assess the similarity between predicted molecules and the true molecule using two molecular similarity measures. First, we use the maximum common edge subgraph (MCES) metric [43], which is an edit distance on molecular graphs. Specifically, we evaluate how close the most similar prediction is to the true molecule in terms of the MCES distance across top-$k$ predictions (we evaluate $k \in \{1, 10\}$):

$$\text{Top-}k \text{ MCES: } \min_{\hat{G} \in \hat{\mathcal{G}}_k} MCES(\hat{G}, G), \tag{2}$$

averaged across test examples. The MCES distance is 0 when two graphs are identical, and increasing values correspond to decreasing similarity. We also use the Tanimoto similarity (or Jaccard coefficient) on the Morgan fingerprints of molecules [53], which measures how well a generative model recognizes true molecular fragments:

$$\text{Top-}k \text{ Tanimoto: } \max_{\hat{G} \in \hat{\mathcal{G}}_k} Tanimoto(\hat{G}, G). \tag{3}$$

The Tanimoto similarity between two molecules ranges from 0 to 1, where a value of 1 indicates identical molecules.

**Molecule retrieval.** In practice, *de novo* molecule generation is often infeasible due to the combinatorial complexity of the solution space. An alternative and practically relevant problem is molecule retrieval, which is to rank candidate molecular graphs (from a chemical database) for a given input spectrum. Formally, given a mass spectrum $X \subset \mathbb{R}_+ \times (0, 1]$, the task is to order a set of candidate graphs $\mathcal{C} = \{G_1, \dots, G_n\}$ such that the correct molecular graph $G \in \mathcal{C}$ has the lowest index.

Chemical databases may contain millions of molecules, e.g., the PubChem database has over 118 million molecules [54], making it impractical to sort the entire set. However, since the mass of the true molecule can be derived from an MS/MS spectrum, the candidate set $\mathcal{C}$ can be constructed to include only molecules with same masses as the true one (within an acceptable experimental error range). To standardize the task across examples, we limit $|C| \leq 256$ candidates per spectrum, sampled randomly if more molecules with the same mass are available. Additionally, similar to the *de novo* generation task, we define a bonus challenge where the set $V$ is known via the molecular formula, allowing further pruning of $\mathcal{C}$ to include only graphs with the given nodes $V$.

We evaluate molecule retrieval using standard information retrieval metrics, as well as the molecular similarity of the top hit with the true molecule. Specifically, we measure:

$$\text{Hit rate @ } k: \mathbb{1}\{G \in \mathcal{C}_k\}, \tag{4}$$

averaged across all test examples, where $\mathcal{C}_k \subset \mathcal{C}$ is the set of top-$k$ hits sorted by the model and $\mathbb{1}$ is the indicator function. The hit rate @ $k$ ranges from 0 to 1, with 1 indicating perfect performance, meaning all true molecules were correctly retrieved among the top $k$ candidates. Additionally, we evaluate the average similarity of the top-1 hit $G_1 \in \mathcal{C}$ with the ground truth molecule $G$ by measuring the maximum common edge subgraph (MCES) distance [43]:

$$\text{MCES @ } 1: MCES(G_1, G). \tag{5}$$

The MCES @ 1 value is 0 if the top-1 retrieved candidate is exactly the true molecule, with higher positive values indicating lower similarity between the molecules.

**Spectrum simulation.** In contrast to the above spectrum-to-molecule tasks, spectrum simulation is the inverse problem of predicting an MS/MS spectrum. The input is a molecular graph $G = (V, E)$ and the measurement parameters $I \in \{I_1, I_2, \dots, I_n\}$, $A \in \{A_1, A_2, \dots, A_m\}$, and $C \in \mathbb{R}_+$, where $I$ represents the type of instrument used, $A$ represents the adduct associated with the precursor ion, and $C$ is the amount of energy used during fragmentation (the collision energy, measured in electronvolts or eV). The output is a predicted mass spectrum $\hat{X} \subset \mathbb{R}_+ \times (0, 1]$. To limit complexity from extra parameters tangential to the issue, we restrict the task to spectra with the most abundant adduct ($A = [\text{M+H}]^+$) and simplify the instrument types to the two principal technologies ($I \in \{\text{QTOF, Orbitrap}\}$).

Typically, $\hat{X}$ and the true spectrum $X$ have binned representations $\mathbf{x}, \hat{\mathbf{x}} \in \mathbb{R}^d$, where $d$ is the number of bins. Instead of listing exact locations of m/z peaks and their intensities, they discretize the space

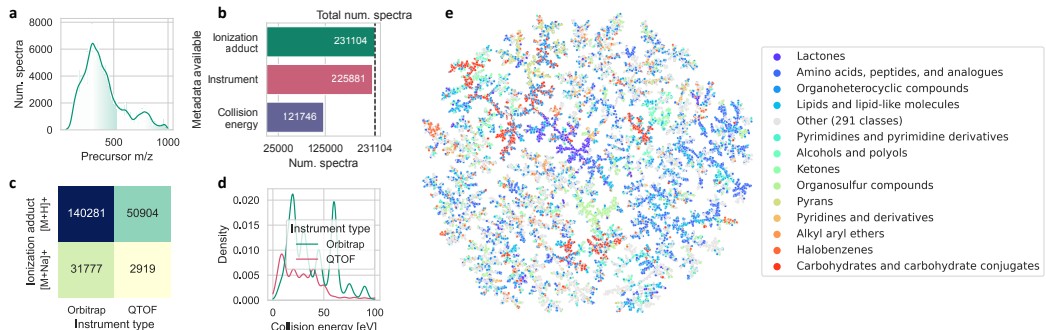

Figure 3: **MassSpecGym provides a diverse and highly-standardized dataset of MS/MS spectra.**
The histogram of precursor m/z values (**a**) and a TMAP [58] projection of precursor molecules (**e**)
demonstrate a rich coverage of molecular masses and chemical classes [44] in MassSpecGym. Unlike
other spectral libraries, our dataset is highly standardized in terms of mass spectrometry metadata.
Each spectrum has an associated ionization adduct, either [M+H]+ or [M+Na]+, and nearly all spectra
(98%) are linked to MS instruments, either Orbitrap or QTOF (**b, c**). Approximately half of the
dataset entries (53%) contain normalized collision energies (**b, d**).

into a series of m/z bins to store peaks in their approximate positions [55, 56, 57]. The selection of
bin size, and by extension $d$, requires consideration: in this benchmark, we choose a bin size of 0.01
Da and a maximum m/z of 1005 Da, resulting in $d = 100500$. These values are precise enough to
retain important information about peak accuracy without becoming overly sensitive to measurement
error. Additionally, we exclude potential precursor signals from both ground truths and predictions in
the benchmark since there is a tendency for strong precursor signals to inflate model performance.

An evaluation metric for the quality of the prediction is the cosine similarity between the predicted
binned spectrum $\hat{\mathbf{x}}$ and the ground truth $\mathbf{x}$ (Equation 6), averaged across all graph-spectra pairs:

$$\text{Cosine similarity:} \frac{\hat{\mathbf{x}}^T \mathbf{x}}{\|\hat{\mathbf{x}}\|\|\mathbf{x}\|}. \tag{6}$$

Cosine similarity between two spectra ranges from 0 to 1, where 1 corresponds to a perfect prediction.
Jensen-Shannon similarity is reported as an additional metric (see Supplementary Information).

A key application of accurately predicting spectra from molecules is in molecular retrieval [55, 21, 23].
Accurate and scalable models enable the automatic annotation of molecular databases, bolstering the
coverage of existing spectral libraries. Therefore, we can additionally use an analagous setup as a
downstream task to evaluate spectrum predictions. Similarly to the molecule retrieval task defined
previously, for a molecule-spectrum pair $(G, X)$, the set of candidates $\mathcal{C}$ comprises of $G$ and the set
of molecules from a chemical database most similar to $G$. For each molecule $G_i \in \mathcal{C}$, we predict a
spectrum $\hat{\mathbf{x}}_i$ and rank all candidates by decreasing cosine similarity between $\hat{\mathbf{x}}_i$ and $\mathbf{x}$. We evaluate
the model by the rate at which $G$ is ranked in the top-$k$ hits in the sorted $\mathcal{C}$, using the same hit rate @
$k$ metrics as defined in the previous section.

### 3.3 Dataset collection

To construct the MassSpecGym dataset, we first exhaustively collected MS/MS spectra from the
largest publicly available spectral libraries: MoNA [37], MassBank [38], and GNPS [36] (downloaded
from the official websites on May 27, 2024), as well as from our in-house data [34]. We then
deduplicated and cleaned the spectra by applying a series of matchms filtering criteria [59]. These
criteria are mainly based on the protocol described in [60] and involve additional filters to better
standardize the dataset, such as keeping only spectra of molecules with m/z < 1000 or spectra
with fewer than 1000 signals. To ensure the high quality of the dataset, we applied additional
criteria, such as removing all spectra where more than 50% of the total intensity cannot be explained
by combinatorially decomposing molecular mass into plausible chemical subformulae [61]. We
preprocessed the mass spectra by removing signals estimated to be instrument noise. Finally, we
standardized both molecular structures and mass spectra, and harmonized metadata entries, inferring

missing or incorrect values where possible [62, 60]. Figure 3 shows that our resultant dataset is rich in terms of molecular structures and highly standardized in terms of mass spectrometry metadata.

Additionally, we provide curated unlabeled datasets of mass spectra and molecules. For the mass spectra, we provide the GeMS-A10 dataset, a deduplicated collection of 24 million high-quality mass spectra mined from the MassIVE repository [20]. For the molecules, we provide (i) 1 million molecules of biological and environmental origin, including collections of natural products, pesticides, industrial chemicals, food additives, and other compounds [43], (ii) 4 million molecules spanning a diverse range of chemical classes [35], and (iii) all 118 million molecules from PubChem [63] (downloaded from the official website on May 31, 2024).

First, we utilize these three molecular datasets to construct retrieval candidates $\mathcal{C}$ for the molecule retrieval and spectrum simulation tasks (Section 3.2). For each spectrum-molecule pair, we iteratively sample molecules with the same mass as the query molecule from (i), followed by (ii) and (iii) until the maximum number of candidates $|\mathcal{C}| = 256$ is reached. The sequence of the datasets used for sampling reflects the relevance of their composition for mass spectrometry applications. A similar procedure is applied for the bonus challenge, where candidates are selected based on identical molecular formulae.

Second, when developing new methods that leverage unlabeled data, we anticipate that users will rely solely on the following two datasets: the GeMS-A10 dataset of unlabeled MS/MS spectra and a refined subset of the unlabeled 4 million-molecule dataset. The molecular dataset has been refined by excluding any molecules with an MCES distance of less than two from any molecule in the test fold of MassSpecGym. This refinement is intended to reduce the potential for data leakage, particularly when used in the context of the *de novo* generation challenge.

## 3.4 Dataset splitting

We split our dataset using MCES distances between molecular graphs corresponding to mass spectra. Specifically, we group all 29 thousand unique molecules into training, validation, and test folds using agglomerative clustering with MCES as the metric and the minimum distance as the linkage criterion. By setting the linkage distance threshold to 10, our approach ensures that no molecules have an edge edit distance of less than 10 between different data folds. Figure 2 shows that our method significantly surpasses the commonly applied 2D InChIKey disjoint approach, used in nearly all related works, in terms of preventing data leakage. Additionally, we stratify the spectra by instrument types, collision energies, ionization adducts, and the frequency of the molecules in the entire dataset, resulting in balanced folds with respect to metadata. A more detailed description of the splitting and additional analysis is available in the Supplementary Information.

# 4 Experiments

## 4.1 Baseline models

To establish reference performance across the tasks, we evaluate an initial set of representative baseline methods summarized in this section. Please see Supplementary Information for details.

*De novo* **molecule generation.** For the *de novo* molecule generation challenge, we begin by implementing a baseline model based on prior chemical knowledge, referred to as **Random chemical generation**, which produces random chemically valid molecules given specific molecular masses or formulae. This baseline uses combinatorial and graph theory algorithms, drawing from statistics derived from the training data. To complement this domain-knowledge baseline, we also implement two Transformer models [64]. These models encode two-dimensional continuous tokens, representing m/z–intensity value pairs of MS/MS spectra, and decode string representations of molecular graphs. The first model, **SMILES Transformer**, decodes molecules as byte-pair-encoded [65] SMILES strings [66]. The second model, **SELFIES Transformer**, decodes molecules as SELFIES strings [67], offering the advantage of always producing valid chemical structures. We do not include any published state-of-the-art baselines because, to the best of our knowledge, all are either not publicly available or leverage proprietary data for training [24, 68, 69, 70].

Table 2: **Baseline results for the *de novo* molecule generation challenge.** The values in brackets indicate 99.9% confidence intervals upon bootstrapping (20,000 resamples).

| | Top-1 | | | Top-10 | | |
|---|---|---|---|---|---|---|
| | Accuracy ↑ | MCES ↓ | Tanimoto ↑ | Accuracy ↑ | MCES ↓ | Tanimoto ↑ |
| Random chemical generation | 0.00 | **28.59 (28.33-28.84)** | 0.07 (0.07 - 0.07) | 0.00 | 25.72 (25.49-25.95) | 0.10 (0.10 - 0.10) |
| SMILES Transformer | 0.00 | 53.80 (52.95-54.61) | 0.07 (0.07 - 0.08) | 0.00 | 21.97 (21.79-22.16) | **0.17 (0.17 - 0.17)** |
| SELFIES Transformer | 0.00 | 33.28 (33.00-33.57) | **0.10 (0.10 - 0.10)** | 0.00 | **21.84 (21.67-22.00)** | 0.15 (0.15 - 0.15) |
| *Bonus chemical formulae challenge* | | | | | | |
| SMILES Transformer | 0.00 | 79.39 (78.64-80.08) | 0.03 (0.03 - 0.04) | 0.00 | 52.13 (51.45-52.81) | 0.10 (0.09 - 0.10) |
| SELFIES Transformer | 0.00 | 38.88 (38.57-39.20) | **0.08 (0.08 - 0.08)** | 0.00 | 26.87 (26.66-27.11) | **0.13 (0.13 - 0.13)** |
| Random chemical generation | 0.00 | **21.11 (20.97-21.26)** | **0.08 (0.08 - 0.08)** | 0.00 | **18.25 (18.14-18.35)** | 0.11 (0.11 - 0.11) |

Table 3: **Baseline results for the molecule retrieval challenge.** The values in brackets indicate 99.9% confidence intervals upon bootstrapping (20,000 resamples).

| | Hit rate @ 1 ↑ | Hit rate @ 5 ↑ | Hit rate @ 20 ↑ | MCES @ 1 ↓ |
|---|---|---|---|---|
| Random | 0.37 (0.24-0.54) | 2.01 (1.68-2.39) | 8.22 (7.53-8.89) | 30.81 (30.40-31.21) |
| DeepSets | 1.47 (1.18-1.77) | 6.21 (5.64-6.82) | 19.23 (18.24-20.26) | 25.11 (24.84-25.39) |
| Fingerprint FFN | 2.54 (2.17-2.99) | 7.59 (6.96-8.28) | 20.00 (19.01-20.98) | 24.66 (24.38-24.94) |
| DeepSets + Fourier features | 5.24 (4.71-5.83) | 12.58 (11.80-13.42) | 28.21 (27.10-29.36) | 22.13 (21.85-22.43) |
| MIST | **14.64 (13.82-15.54)** | **34.87 (33.69-36.10)** | **59.15 (57.89-60.39)** | **15.37 (15.12-15.62)** |
| *Bonus chemical formulae challenge* | | | | |
| Random | 3.06 (2.64-3.52) | 11.35 (10.60-12.12) | 27.74 (26.52-28.84) | 13.87 (13.70-14.03) |
| DeepSets | 4.42 (3.92-4.97) | 14.46 (13.58-15.36) | 30.76 (29.67-31.93) | 15.04 (14.89-15.19) |
| Fingerprint FFN | 5.09 (4.57-5.66) | 14.69 (13.83-15.56) | 31.97 (30.86-33.10) | 14.94 (14.79-15.09) |
| DeepSets + Fourier features | 6.56 (5.95-7.16) | 16.46 (15.58-17.35) | 33.46 (32.39-34.59) | 14.14 (13.98-14.31) |
| MIST | **9.57 (8.88-10.30)** | **22.11 (21.10-23.13)** | **41.12 (39.98-42.34)** | **12.75 (12.59-12.91)** |

Table 4: **Baseline results for the spectrum simulation challenge.** The values in brackets indicate 99.9% confidence intervals upon bootstrapping (20,000 resamples).

| | Cosine Similarity ↑ | Jensen-Shannon Similarity ↑ | Hit Rate @ 1 ↑ | Hit Rate @ 5 ↑ | Hit Rate @ 20 ↑ |
|---|---|---|---|---|---|
| Precursor m/z | 0.15 (0.14-0.17) | 0.59 (0.58-0.60) | 0.38 (0.21-0.62) | 1.72 (1.32-2.18) | 7.17 (6.32-8.04) |
| FFN Fingerprint | 0.25 (0.24-0.26) | 0.69 (0.63-0.65) | 8.44 (7.56-9.34) | 21.43 (20.10-22.79) | 38.57 (36.99-40.23) |
| GNN | 0.19 (0.18-0.20) | 0.64 (0.63-0.65) | 3.95 (3.37-4.62) | 11.92 (10.87-13.00) | 26.27 (24.83-27.82) |
| FraGNNet | **0.52 (0.51-0.53)** | **0.91 (0.91-0.92)** | **46.64 (44.98-48.26)** | **72.56 (71.18-74.00)** | **83.58 (82.34-84.75)** |
| *Bonus chemical formulae challenge* | | | | | |
| Precursor m/z | - | - | 2.09 (1.66-2.59) | 8.52 (7.65-9.53) | 22.65 (21.26-24.01) |
| FFN Fingerprint | - | - | 7.62 (6.77-8.54) | 22.70 (21.32-24.12) | 44.12 (42.51-45.75) |
| GNN | - | - | 3.63 (3.05-4.29) | 13.55 (12.46-14.68) | 33.77 (32.26-35.37) |
| FraGNNet | - | - | **31.93 (30.40-33.50)** | **63.20 (61.64-64.76)** | **82.70 (81.45-83.93)** |

**Molecule retrieval.** The simplest baseline method for molecule retrieval, **Random**, sorts the candidate molecules $\mathcal{C}$ randomly. The second method, **Fingerprint FFN**, employs a feedforward neural network to predict the Morgan fingerprint of the target molecule. The candidates are then sorted based on their cosine similarity to the predicted fingerprint. Next, we evaluate **MIST**, a state-of-the-art deep learning approach, also based on fingerprint prediction. MIST assigns chemical subformulae to spectral peaks via energy-based modeling [22], then predicts a molecular fingerprint via a chemical formula-based transformer, and finally ranks the candidates by cosine similarity between the fingerprints [19]. Finally, we evaluate **DeepSets** [71]. The model processes spectra as sets of raw 2D peak representations, providing a complementary approach to FingerprintFFN and the state-of-the-art MIST which are based on alternative representations of spectra. **DeepSets + Fourier features** enhances DeepSets by using Fourier features enabling more accurate modeling of m/z values [20].

**Spectrum simulation.** We include three deep learning baseline models for the spectrum simulation task. The **molecular fingerprint (FFN Fingerprint)** model consists of a simple feedforward network on top of a fingerprint representation of the input molecule, inspired by [55]. The **graph neural network (GNN)** model, inspired by [56, 72], uses a variant of Graph Isomorphism Network augmented with edge features [73, 74] to process a 2D graph representation of the input molecule. Finally, state-of-the-art **FraGNNet** [23] uses combinatorial fragmentation and GNNs to parametrize

a probability distribution over fragments of the input molecule and their associated chemical formulae. The precise formula masses are used to map the distribution over formulae to a high resolution mass spectrum, without requiring binning. In addition, we include a trivial baseline **Precursor m/z** that simply predicts a single-peak spectrum by calculating the precursor m/z from the masses of the input molecule and the ionization adduct.

## 4.2 Baseline performance

We train and validate the performance of baseline methods on MassSpecGym. For the challenge of *de novo* generation (Table 2), we find that none of the baselines achieve an accuracy above zero, emphasizing the need for new method development. Additionally, our SMILES Transformer baseline does not outperform random generation of chemically valid graphs, highlighting the insufficiency of simplistic learning approaches in our generalization-demanding setup. For molecule retrieval (Table 3), the advanced MIST model significantly outperforms the simpler Fingerprint FFN and DeepSets baselines, suggesting strong gains from algorithmic development, which we posit as a driving force for MS/MS annotation with our benchmark. The same holds true for the spectrum simulation challenge (Table 4), where the advanced FraGNNet model demonstrates superior performance over simpler baselines. Nevertheless, the absolute metric scores still leave a substantial gap for future improvements.

## 5 Conclusions

In this work, we developed MassSpecGym, the first comprehensive and standardized benchmark for the discovery and identification of molecules from MS/MS spectra. MassSpecGym is based on our newly created largest open-source dataset of labeled tandem mass spectra and a standardization pipeline ensuring high data quality. We split the dataset using our novel generalization-demanding splitting technique, enabling robust evaluation of molecular identification and discovery. We evaluated a series of baseline methods and demonstrated that the annotation of MS/MS spectra remains a highly unsolved problem. To address this, we provide MassSpecGym as a public resource with a user-friendly interface requiring minimal domain expertise for the submission and evaluation of new machine learning models.

Our future work has two main directions. First, we plan to continuously update MassSpecGym with new public and in-house MS/MS data, potentially incorporating simulated spectra or additional data modalities, such as EI spectra. We also aim to expand the scope of challenges to include tasks such as molecular networking, a prominent technique in the field that focuses on clustering spectra of structurally related molecules rather than predicting individual molecules. Second, by progressively enhancing the MassSpecGym ecosystem with more advanced methods, we intend to transform it into a hub for state-of-the-art models in MS/MS spectra annotation. This will empower machine learning researchers to make rapid progress in developing innovative models, a particularly crucial focus given the historically limited collaboration between mass spectrometry experts and AI specialists. As a consequence, many well-established machine learning paradigms, such as generating molecular graphs via diffusion models or applying domain adaptation techniques across different mass spectrometry systems, remain largely unexplored. Furthermore, by providing a user-friendly interface to run these models, we aim to make cutting-edge algorithms readily accessible to life scientists interested in annotating their mass spectra. We believe that MassSpecGym will play a pivotal role in fostering the development of next-generation machine learning methods, ultimately driving significant progress across the biomedical and chemical sciences.

## Acknowledgments and Disclosure of Funding

The idea of this benchmark set was conceived at the Dagstuhl Seminar #24181 "Computational Metabolomics: Towards Molecules, Models, and their Meaning". We are grateful for the funding and support provided by the Leibniz Center for Informatics.

This work was supported by the Ministry of Education, Youth and Sports of the Czech Republic through the e-INFRA CZ (ID:90140) and by the Technology Agency of the Czech Republic through the project RETEMED (TN02000122). This work was also co-funded by the European Union (ERC FRONTIER No. 101097822; ELIAS No. 101120237). Views and opinions expressed are however

those of the author(s) only and do not necessarily reflect those of the European Union or the European Research Council. Neither the European Union nor the granting authority can be held responsible for them. SH and AK are funded through Award R35GM148219 by the NIGMS of the National Institutes of Health. TP was supported by the Czech Science Foundation (GA CR) grant 21-11563M and by the European Union's Horizon 2020 research and innovation programme under Marie Skłodowska-Curie grant agreement No. 891397. AY was supported by a Natural Sciences and Engineering Research Council of Canada (NSERC) scholarship and a Vector Institute research grant. LZ was supported by NSERC. FW was supported by Alberta Machine Intelligence Institute (AMII), and NSERC grant. RG was was supported by NSERC, AMII. DSW was supported by NSERC, and Genome Canada, Genome British Columbia and Genome Alberta. HR was supported by NSERC and the Canada Research Chair (CRC) Program. FK, KD and SB are supported by Deutsche Forschungsgemeinschaft (BO 1910/23). FK and SB are supported by the Ministry for Economics, Sciences and Digital Society of Thuringia (Framework ProDigital, DigLeben-5575/10-9). NAH and SB are supported by the Thüringer Ministerium für Wirtschaft, Wissenschaft und Digitale Gesellschaft (TMWWDG) with funds from the European Union as part of the European Regional Development Fund (ERDF, 2023 VFE 0003). JH and WB acknowledge support by the University of Antwerp Research Fund. FH was supported by Deutsche Forschungsgemeinschaft (DFG, 528775510). LL was supported by NSF Award 2239869. CB was supported by the Czech Academy of Sciences PPLZ fellowship number L200552251.

## Competing interests

RS and TP are co-founders of the company mzio GmbH, which develops technologies related to mass spectrometry data processing. SB, KD and ML are co-founders of Bright Giant GmbH. JJJvdH is member of the Scientific Advisory Board of NAICONS Srl., Milano, Italy and consults for Corteva Agriscience, Indianapolis, IN, USA.

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
