# MassSpecGym: Supplementary information

**Roman Bushuiev**[1,2]**, Anton Bushuiev**[2]**, Niek F. de Jonge**[3]**, Adamo Young**[4]**,**
**Fleming Kretschmer**[5]**, Raman Samusevich**[1,2]**, Janne Heirman**[6]**, Fei Wang**[7,8]**,**
**Luke Zhang**[9]**, Kai Dührkop**[5]**, Marcus Ludwig**[10]**, Nils A. Haupt**[5]**, Apurva Kalia**[11]**,**
**Corinna Brungs**[1]**, Robin Schmid**[1]**, Russell Greiner**[7,8]**, Bo Wang**[4]**, David S. Wishart**[7,12]**,**
**Li-Ping Liu**[11]**, Juho Rousu**[13]**, Wout Bittremieux**[6]**, Hannes Rost**[9]**, Tytus D. Mak**[14]**,**
**Soha Hassoun**[11,15]**, Florian Huber**[16]**, Justin J.J. van der Hooft**[3,17]**, Michael A. Stravs**[18]**,**
**Sebastian Böcker**[5]**, Josef Sivic**[2]**, Tomáš Pluskal**[1]

[1]Institute of Organic Chemistry and Biochemistry of the Czech Academy of Sciences,
[2]Czech Institute of Informatics, Robotics and Cybernetics, Czech Technical University,
[3]Bioinformatics Group, Wageningen University & Research, [4]Department of Computer
Science, University of Toronto, [5]Chair for Bioinformatics, Institute for Computer Science,
Friedrich Schiller University Jena, [6]Department of Computer Science, University of Antwerp,
[7]Department of computing science, University of Alberta, [8]Alberta Machine Intelligence Institute,
[9]Department of Molecular Genetics, University of Toronto, [10]Bright Giant GmbH, [11]Department of
Computer Science, Tufts University, [12]Department of Biological Sciences, University of Alberta,
[13]Department of Computer Science, Aalto University, [14]Mass Spectrometry Data Center,
National Institute of Standards and Technology, [15]Department of Chemical and Biological
Engineering, Tufts University, [16]Centre for Digitalisation and Digitality, University of Applied
Sciences Düsseldorf, [17]Department of Biochemistry, University of Johannesburg,
[18]Eawag: Swiss Federal Institute of Aquatic Science and Technology

## Contents

# 1 Dataset and code access

## 1.1 Availability

Following the NeurIPS Dataset and Benchmark Track guidelines, we have made our dataset publicly available under the MIT license. The dataset and its Croissant metadata record can be accessed through the Hugging Face Hub. Furthermore, we have made the code for dataset construction, analysis, reproduction of all our experiments, and evaluation of new models available on GitHub under the MIT license. Figure 1 provides an overview of the MassSpecGym infrastructure. We bear all responsibility in case of any violation of rights.

- MassSpecGym dataset:
  `https://huggingface.co/datasets/roman-bushuiev/MassSpecGym`.
- MassSpecGym code:
  `https://github.com/pluskal-lab/MassSpecGym`.

## 1.2 Variable list

Table 1 outlines the structure of our MassSpecGym dataset and provides the list of all variables along with their descriptions. Figure 2 explains the key relationship between the main variables of individual samples: multiple spectra with different metadata, measured under varying experimental setups, may be annotated with the same molecule.

## 1.3 Dataset Applications

The primary goal of MassSpecGym is to identify the most effective machine learning models for the annotation of MS/MS spectra with molecular structures. Our benchmark, which defines a generalization-demanding data split and practically-motivated evaluation metrics, ensures that models performing well on MassSpecGym can be effectively applied to real-world, unannotated data. Ultimately, MassSpecGym paves the way for new biological and chemical discoveries by stimulating advances in the annotation of MS/MS spectra. On the other hand, *de novo* molecule generation serves

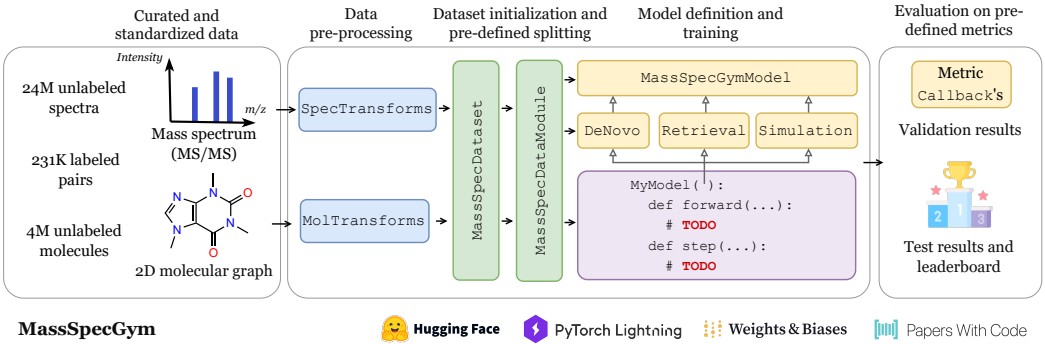

Figure 1: **MassSpecGym enables a standardized and user-friendly evaluation of machine learning methods for MS/MS annotation via an easily extendable modular interface.** The dataset can be loaded, preprocessed, split, and utilized for training, evaluation, and metric logging through the prepared codebase. To develop and evaluate a new model, a user only needs to implement a forward pass with custom prediction logic. Colored blocks represent classes in our codebase (https://github.com/pluskal-lab/MassSpecGym). Arrows with empty heads represent subclass inheritance, while arrows with bold heads conceptually show the flow from the dataset to the evaluation metrics.

Table 1: **Overview of all variables present in the MassSpecGym dataset.** $n$ denotes the number of signals in a spectrum. Floating point variables were rounded to four decimal places for computing the number of unique entries. The key variables in the MassSpecGym dataset are `mzs` and `intensities` representing an input spectrum, and `smiles`, representing the target molecule.

| Variable | Description | Data type | Num. unique values | Example |
|---|---|---|---|---|
| `identifier` | Unique entry identifier | string | 231,104 | MassSpecGymID0088683 |
| `mzs` | Array of spectrum m/z values | $n \times$ float | 231,104 | [55.0542, 57.0699, . . . , 238.0995] |
| `intensities` | Array of spectrum intensities | $n \times$ float | 231,104 | [0.0240, 1.0, . . . , 0.5356] |
| `smiles` | SMILES string of molecule | string | 31,602 | CCCCOCN(C1=C(C=C. . . CCl |
| `inchikey` | 2D InChI key | string | 28,929 | HKPHPIREJKHECO |
| `formula` | Chemical formula of molecule | string | 17,634 | C17H26ClNO2 |
| `precursor_formula` | Chemical formula of precursor ion | string | 21,653 | C17H27ClNO2 |
| `parent mass` | Mass of molecule | float | 32,228 | 311.1652 |
| `precursor_mz` | M/z of precursor ion | float | 32,275 | 312.1725 |
| `adduct` | Ionization adduct | string | 2 | [M+H]+ |
| `instrument_type` | Type of MS instrument | string | 2 | Orbitrap |
| `collision_energy` | Energy of CID fragmentation | float | 9,737 | 30.0 |
| `fold` | Split fold which entry belongs to | string | 3 | train |
| `simulation_challenge` | Entry is used for simulation challenge | boolean | 2 | True |

as a benchmark for evaluating novel generative modeling algorithms. With the rapid advancements in the field of deep generative modeling, our benchmark provides a rigorous case study to assess their capabilities. Similarly, the molecule retrieval challenge acts as a benchmark for evaluating information retrieval algorithms. The spectrum simulation benchmark has the potential to provide insights into the mechanisms of MS/MS fragmentation, thereby deepening our understanding of the underlying processes in analytical chemistry.

## 1.4 Target Audiences

MassSpecGym aims to democratize and popularize the challenge of annotating molecular structures from MS/MS spectra. Our platform targets two primary audiences:

- Machine learning community: Researchers and developers specializing in creating new machine learning algorithms, who may not necessarily have a background in mass spectrometry or other fields of chemistry.

- Metabolomics community: Scientists and professionals specializing in mass spectrometry-related fields, such as natural product chemists, who can utilize models trained on MassSpecGym and apply insights from the MassSpecGym results to their research problems.

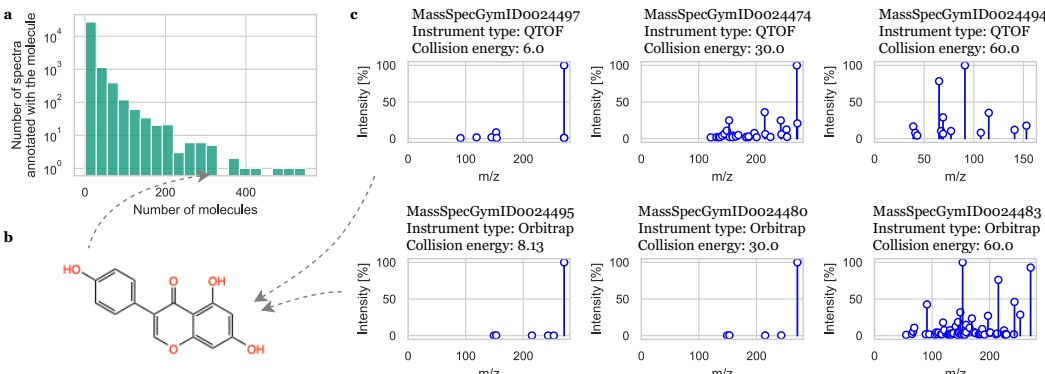

Figure 2: **Same measured molecule may result in different MS/MS spectra under different mass spectrometry measurement conditions. a,** The distribution of the number of spectra corresponding to the same molecule in the MassSpecGym dataset. **b,** Example of a molecule annotating 321 spectra in the dataset. **c,** Examples of six spectra annotated with the molecule shown in figure **b**: different instrument types and collision energies lead to different spectra. Higher collision energies typically lead to richer fragmentation of a molecule, resulting in a higher number of peaks in the spectrum.

MassSpecGym streamlines the cleaning and pre-processing of data, removing the necessity for specialized mass spectrometry expertise. This makes the platform accessible to a wider audience, including those who may not have access to extensive computational resources for collecting and cleaning large-scale data (e.g., large unlabeled datasets provided in our benchmark).

## 2  MassSpecGym benchmark construction

This section provides details on the construction of the MassSpecGym benchmark that are not covered in the main text.

### 2.1  Definition of challenges

**Morgan fingerprint.**  A Morgan fingerprint [1] is a set of structural features of a molecule. For machine learning purposes, it is typically represented as a bit vector, where each bit indicates the presence of a specific fragment in the molecule. The dimensionality of a Morgan fingerprint refers to the number of bits in the vector, usually set to a fixed size such as 2048 or 4096 bits. An additional parameter of the Morgan fingerprint is its radius. The radius specifies the number of steps (or bonds) to consider from each atom when generating the fingerprint, with a common choice being a radius of 2, which considers all the neighbors up to two bonds away for each atom. This helps capture the local structural environment around each atom within the molecule. Further, we denote a Morgan fingerprint of a molecular graph $G$, representing a sets of it structural features, as $fingerprint(G)$.

**Tanimoto Similarity**  Tanimoto similarity measures the similarity between two molecular graphs $G_1, G_2$ using their Morgan fingerprints. It is defined as the intersection over union of the two corresponding fingerprints:

$$Tanimoto(G_1, G_2) = \frac{|fingerprint(G_1) \cap fingerprint(G_2)|}{|fingerprint(G_1) \cup fingerprint(G_2)|}. \qquad (1)$$

The Tanimoto similarity score ranges from 0 to 1, with 1 indicating identical structures. However, in rare cases, non-identical structures can also produce a similarity score of 1 due to fingerprint collisions. A similarity value above 0.85 has been previously used in several studies as a threshold for identifying compounds with similar activity [2].

**MCES distance.**  To measure similarity between two molecular structures, additional to the Tanimoto similarity, the Maximum Common Edge Subgraph (MCES) distance is used on the molecular graphs [3]. MCES distance can be understood as an edit distance on graphs, representing the minimum

number of edges that have to be removed for both graphs to be isomorphic, ignoring singleton nodes. This measure of molecular similarity provides better interpretability than the Tanimoto similarity and more accurately reflects the biochemical modifications of molecules. Formally, given two graphs $G_1 = (V_1, E_1)$ and $G_2 = (V_2, E_2)$, where $V_1, V_2$ are the sets of vertices and $E_1, E_2$ are the sets of edges, the maximum common edge subgraph $G_c = (V_c, E_c)$, $V_c \subseteq V_i$, $E_c \subseteq E_i$ for both $i \in \{1, 2\}$ is a graph that maximizes $|E_c|$. The MCES distance is then defined as

$$MCES(G_1, G_2) = |E_1| + |E_2| - 2|E_c|, \tag{2}$$

which is minimized by the maximum common edge subgraph. We use the myopicMCES implementation (`https://github.com/AlBi-HHU/myopic-mces`), which computes the exact distance below a specified threshold and a guaranteed lower bound above; the distance is weighted by bond order. Unless otherwise stated, we use a threshold of 15 and enable the `always_stronger_bound` option (default) which always computes a stronger lower bound first despite higher computational demand.

## 2.2 Data collection

**Public repositories.** The mass spectral data is composed of various LC-MS/MS databases and datasets, which were downloaded on May 27th, 2024, from:

- MoNA [4] downloaded from https://mona.fiehnlab.ucdavis.edu/downloads
- Massbank [5] downloaded from https://github.com/MassBank/MassBank-data/releases
- GNPS [6] The following libraries were downloaded from https://gnps-external.ucsd.edu/gnpslibrary
    - BERKELEY-LAB.mgf
    - BILELIB19.mgf
    - BIRMINGHAM-UHPLC-MS-NEG.mgf
    - BIRMINGHAM-UHPLC-MS-POS.mgf
    - BMDMS-NP.mgf
    - CASMI.mgf
    - DRUGS-OF-ABUSE-LIBRARY.mgf
    - ECG-ACYL-AMIDES-C4-C24-LIBRARY.mgf
    - ECG-ACYL-ESTERS-C4-C24-LIBRARY.mgf
    - GNPS-COLLECTIONS-MISC.mgf
    - GNPS-COLLECTIONS-PESTICIDES-NEGATIVE.mgf
    - GNPS-COLLECTIONS-PESTICIDES-POSITIVE.mgf
    - GNPS-D2-AMINO-LIPID-LIBRARY.mgf
    - GNPS-EMBL-MCF.mgf
    - GNPS-FAULKNERLEGACY.mgf
    - GNPS-IOBA-NHC.mgf
    - GNPS-LIBRARY.mgf
    - GNPS-MSMLS.mgf
    - GNPS-NIH-CLINICALCOLLECTION1.mgf
    - GNPS-NIH-CLINICALCOLLECTION2.mgf
    - GNPS-NIH-NATURALPRODUCTSLIBRARY.mgf
    - GNPS-NIH-NATURALPRODUCTSLIBRARY_ROUND2_NEGATIVE.mgf
    - GNPS-NIH-NATURALPRODUCTSLIBRARY_ROUND2_POSITIVE.mgf
    - GNPS-NIH-SMALLMOLECULEPHARMACOLOGICALLYACTIVE.mgf
    - GNPS-NIST14-MATCHES.mgf
    - GNPS-NUTRI-METAB-FEM-NEG.mgf
    - GNPS-NUTRI-METAB-FEM-POS.mgf
    - GNPS-PRESTWICKPHYTOCHEM.mgf
    - GNPS-SAM-SIK-KANG-LEGACY-LIBRARY.mgf
    - GNPS-SCIEX-LIBRARY.mgf

- GNPS-SELLECKCHEM-FDA-PART1.mgf
- GNPS-SELLECKCHEM-FDA-PART2.mgf
- HCE-CELL-LYSATE-LIPIDS.mgf
- HMDB.mgf
- IQAMDB.mgf
- LDB_NEGATIVE.mgf
- LDB_POSITIVE.mgf
- MIADB.mgf
- MMV_NEGATIVE.mgf
- MMV_POSITIVE.mgf
- PNNL-LIPIDS-NEGATIVE.mgf
- PNNL-LIPIDS-POSITIVE.mgf
- PSU-MSMLS.mgf
- RESPECT.mgf
- SUMNER.mgf
- UM-NPDC.mgf

**In-house data.** In-house spectral libraries were acquired for four different compound libraries, including the Bioactive Compound Library and the 5k Scaffold Library from MedChemExpress, the NIH NPAC ACONN collection of Natural Products from NIH/NCATS, and the Alpha-helix Peptidomimetic Library from OTAVAchemicals, resulting in almost 20,000 unique compounds. Up to 10 compounds were pooled in one well of 96 or 384 well plates and diluted to reach a concentration between 5-20 μM. A flow injection method was performed by a Vanquish Duo UHPLC system coupled to an Orbitrap ID-X instrument. 2 μL injection volume was used. The m/z range for $MS^1$ was set to 115 to 2000 with a resolution of 30,000. The three most intense ions were picked by data-dependent acquisition for $MS^2$ experiments with a resolution of 15,000. Detailed information about the data acquisition can be found in [7]. We used the following files downloaded from `https://zenodo.org/records/11163381`.

- 20231031_nihnp_library_neg_all_lib_MSn.mgf
- 20231130_mcescaf_library_neg_all_lib_MSn.mgf
- 20231130_otavapep_library_neg_all_lib_MSn.mgf
- 20240411_mcebio_library_neg_all_lib_MSn.mgf
- 20231031_nihnp_library_pos_all_lib_MSn.mgf
- 20231130_mcescaf_library_pos_all_lib_MSn.mgf
- 20231130_otavapep_library_pos_all_lib_MSn.mgf
- 20240411_mcebio_library_pos_all_lib_MSn.mgf

## 2.3 Data cleaning

To clean the collected dataset, we applied spectrum-based filtering and metadata-based filtering using the matchms package [8], as well as spectral quality-based filtering utilizing SIRIUS software [9] and the NIST20 spectral library [10]. The following subsections describe the individual steps. The reduction in the number of mass spectra after each filtering step is summarized in Table 2, following the matchms cleaning report format.

**Spectrum-based filtering.** Noise was detected and removed by first searching for multiple identical intensity values and then using the associated intensity value multiplied by 2 as a minimum intensity threshold. Spectra that contained more than 300 or less than 1 fragment after noise removal were discarded. Spectra with patterns that suggested that they were stored as profile spectra were also removed. It is not uncommon to depose data in multiple databases; therefore, duplicates, i.e. spectra with identical annotation and a cosine score of 1.0, were removed. We removed all entries with precursor m/z values above 1,000 Da. Additionally, we dropped all spectra containing at least one signal with an m/z value above the precursor m/z value + 3 and an intensity value above 20% of the highest intensity within the spectrum.

**Metadata-based filtering.**  The spectra from public libraries have a high diversity of metadata formats and often have missing or incorrect metadata. The metadata was largely processed using default settings as described in [11] with the full workflow being provided on `https://github.com/pluskal-lab/MassSpecGym/blob/main/notebooks/dataset_construction`.

First all merged spectra in the MS/MS library from Brungs et al. [7] were removed.

Metadata fields were harmonized to have the same field names. Metadata in the wrong field was corrected. Annotations were completed, to have a SMILES, InChI and InChIKey. Missing parent masses were derived from the SMILES mass and missing adducts were derived from the precursor m/z and parent mass. Only spectra where the parent mass calculated from the adduct and precursor m/z matched the given parent mass and SMILES mass were stored.

Annotations were derived from the compound name by searching on PubChem. Spectra were removed if the mono-isomeric mass of the SMILES did not match the parent mass given in the metadata. Annotations which corresponded to a charged molecule were removed. The formula and precursor formula were derived from the SMILES. Additionally, entries with molecules containing rare chemical elements `Sn` and `Al` were removed.

For spectra to be technically comparable we limited the data to LC-MS/MS spectra measured in positive electrospray ionization (ESI) mode, attributable to Orbitrap-type (Orbitrap, ITFT, QFT) or QTOF-type instruments (see below). We further only retained spectra of two adducts, [M+H]+ or [M+Na]+, to reduce additional biases.

Next, we removed all structures that are disconnected (i.e. containing a '.' within the SMILES string). These structures usually belong to salts or metal-containing compounds which are not measured in this form in mass spectrometry. We ended up with 414,049 spectra with proper metadata annotation.

**Quality assessment and filtering.**  When collecting data from a large number of publicly accessible repositories, it is important to consider that some, or even many, of the data might be incorrectly annotated. This issue arises because not all spectra originate from reference measurements; many spectra are annotated via spectral library searches or computational tools before being uploaded to public repositories. In the following analysis, we aim to remove compounds from the training data if there is uncertainty about their correct labeling.

One straightforward check is to count how many peaks, or how much intensity, can be explained by fragment ions whose molecular formulas are subsets of the parent molecule's formula. We use SIRIUS [9] to decompose all peaks in the spectra and remove any spectra from the dataset that explain less than 50% of the total intensity. To prevent a single large, intense peak from dominating the statistics, we apply a square root transformation to all peak intensities (here and in all subsequent filtering steps). We removed 135,190 spectra using this method. Although this number seems high, it only accounts for 926 unique molecular structures. Additionally, we found that the majority of the removed spectra did not contain a single peak that could be explained with a molecular formula subset of the parent molecule.

Next, we compared the spectra against the NIST20 [10] library – a commercially available, high-quality, manually curated spectral library. We found that one third of the structures are contained in both spectral libraries. For 14,600 spectra (38 unique structures), there was not a single peak (beyond the parent peak) shared with the NIST spectra of the same molecular structure. For an additional 9,290 spectra, the maximum cosine similarity with any NIST spectrum of the same structure was below 0.2, or there were fewer than 5 shared peaks. We removed these spectra; again, the number of removed structures was much lower, with only 126 structures removed.

Finally, we compared all spectra of the same structure within our dataset pairwise to identify outliers. We only considered structures with at least 4 spectra and removed 11,740 spectra for which the maximum cosine similarity to any other spectra of the same structure was lower than 0.2. We used such a conservative threshold to avoid removing spectra solely because they were measured at a very different collision energy than the other spectra.

In total, we are left with 231,104 spectra. A manual examination of a small random subset of the removed spectra found all of them at least "suspicious," with many clearly being wrongly annotated.

For each of these remaining spectra, we additionally provide fragment peak-molecular formula annotations in the form of fragmentation trees [12, 13, 14]. Each fragmentation tree annotates the

Table 2: **Summary of our data cleaning pipeline.** The rows in the table provide a report on the number of removed spectra, the number of spectra with modified metadata, and the number of spectra that were removed after each filtering step. The filtering steps were applied sequentially from top to bottom, as listed in the table. A horizontal rule separates the matchms filters from the additional filters that were applied beyond matchms.

| Filter | Removed spectra | Changed metadata | Changed mass spectrum |
|---|---|---|---|
| add_parent_mass | 0 | 699,187 | 0 |
| add_precursor_formula | 0 | 464,082 | 0 |
| add_retention_index | 0 | 706,484 | 0 |
| add_retention_time | 0 | 552,948 | 0 |
| clean_adduct | 0 | 8,102 | 0 |
| clean_compound_name | 0 | 104,201 | 0 |
| correct_charge | 0 | 322,993 | 0 |
| derive_adduct_from_name | 0 | 376,859 | 0 |
| derive_annotation_from_compound_name | 0 | 53,716 | 0 |
| derive_formula_from_name | 0 | 47,156 | 0 |
| derive_formula_from_smiles | 0 | 293,157 | 0 |
| derive_inchi_from_smiles | 0 | 42,470 | 0 |
| derive_inchikey_from_inchi | 0 | 413,405 | 0 |
| derive_ionmode | 0 | 137,815 | 0 |
| derive_smiles_from_inchi | 0 | 38,547 | 0 |
| harmonize_instrument_types | 0 | 294,479 | 0 |
| harmonize_undefined_inchi | 0 | 117,746 | 0 |
| harmonize_undefined_inchikey | 0 | 467,179 | 0 |
| harmonize_undefined_smiles | 0 | 114,075 | 0 |
| normalize_intensities | 0 | 0 | 513,299 |
| remove_charged_molecules | 4,776 | 0 | 0 |
| remove_instrument_types | 13,481 | 0 | 0 |
| remove_noise_below_frequent_intensities | 0 | 0 | 64,805 |
| remove_not_ms2_spectra | 628,478 | 0 | 0 |
| repair_adduct_based_on_smiles | 0 | 305,728 | 0 |
| repair_inchi_inchikey_smiles | 0 | 33,319 | 0 |
| repair_not_matching_annotation | 0 | 1,695 | 0 |
| repair_smiles_of_salts | 0 | 6,596 | 0 |
| require_adduct_in_list | 41,049 | 0 | 0 |
| require_correct_ionmode | 153,321 | 0 | 0 |
| require_formula_match_parent_mass | 152 | 0 | 0 |
| require_matching_adduct_precursor_mz_parent_mass | 3,726 | 0 | 0 |
| require_minimum_number_of_peaks | 1,357 | 0 | 0 |
| require_number_of_peaks_below_maximum | 880 | 0 | 0 |
| require_parent_mass_match_smiles | 8,694 | 0 | 0 |
| require_precursor_mz | 7,291 | 0 | 0 |
| require_valid_annotation | 22,036 | 0 | 0 |
| store_relevant_metadata_only | 0 | 449,721 | 0 |
| make_charge_int | 0 | 0 | 0 |
| add_compound_name | 0 | 0 | 0 |
| interpret_pepmass | 0 | 0 | 0 |
| add_precursor_mz | 0 | 0 | 0 |
| require_matching_adduct_and_ionmode | 0 | 0 | 0 |
| Remove salts | 3,819 | 0 | 0 |
| < 50% explained intensity by molecular formula decomposition | 135,190 | 0 | 0 |
| No peaks shared with NIST spectrum | 14,699 | 0 | 0 |
| Cosine similarity against NIST < 0.2 | 9,290 | 0 | 0 |

fragment peaks of the corresponding MS/MS spectrum with sub-formulas of the parent's molecular formula. These fragmentation trees were computed using SIRIUS (version 6.0.4) with default parameter settings.

**Subsetting spectra for the spectrum simulation challenge.** Since instrument type and collision energy are required inputs for the spectrum simulation challenge, we consider only a subset of the MassSpecGym dataset for this challenge. We note that in this dataset, 98% percent of entries contain the [M+H]+ adduct. Therefore, we additionally removed the other 2% of entries with the [M+Na]+ ionization adduct for the simulation challenge. The boolean mask for this subset is stored in the dataset under the variable named `simulation_challenge`.

## 2.4 Data standardization

**Spectrum standardization.** We standardize all MS/MS spectra to have relative intensity values. Specifically, for each MS/MS spectrum, we divide each intensity value by the maximum intensity

within that spectrum. This normalization process ensures that the spectra are comparable regardless of their absolute intensity values, which can vary due to differing MS experimental conditions.

**Instrument standardization.** Instrument type descriptions were standardized based on the fixed vocabulary for instrument types of the MassBank record format (`https://github.com/MassBank/MassBank-web/blob/dev/Documentation/MassBankRecordFormat.md#2.4.2`) as a basis. For records originating from MassBank or having an instrument type description matching a MassBank notation (e.g., `LC-ESI-ITFT`), the instrument type was retained. For all other records, a mapping was manually generated from the unique values for instrument types in GNPS records. Explicitly specified instrument names were assigned to the code for their instrument types. Clearly nonsensical entries (e.g., `ESI-LC-ESI-IT`, which has two ionization mechanism tags) were mapped to the most plausible explanation (here, `LC-ESI-IT`). Some entries are imprecise and therefore ambiguous. In particular, `Orbitrap` may either correspond to `ITFT` or `QFT`. Finally, for the purpose of the final dataset, only the analyzer type is relevant to the type of spectrum observed, and details related to sample introduction (e.g. `LC`) are irrelevant to the measured data. Therefore, in the dataset, only the analyzer type is reported, where `QTOF` represents all instruments based on a quadrupole-time of flight analyzer and `Orbitrap` represents all instruments with an Orbitrap-type analyzer.

**Collision energy standardization.** The provided collision energies were standardized to appropriately handle normalization. Collision energy normalization is way of representing the fragmentation energy that accounts for the mass of the precursor ion. Assuming the precursor $p$ is singly charged (which was the case for the spectra in our dataset, since we removed all spectra whose adducts were not [M+H]+ or [M+Na]+), the non-normalized collision energy $\mathrm{CE}(p)$ can be calculated from the normalized collision energy $\mathrm{NCE}(p)$ and the associated precursor m/z $\mathrm{m}(p)$ using Equation 3.

$$\mathrm{CE}(p) = \frac{\mathrm{m}(p) \times \mathrm{NCE}(p)}{500} \tag{3}$$

This transformation was applied to all collision energies that were identified as normalized. In cases of ambiguity, we assumed that the provided collision energy was not normalized.

**Molecular structure standardization.** Finally, the molecular structures of all data sets were standardized with respect to their corresponding SMILES strings in order to have a unified and unique representation of compound structures. SMILES were standardized using the PubChem standardization service accessed through the standardizeUtils python package (`https://github.com/boecker-lab/standardizeUtils`), which utilizes the PubChem Power User Gateway (PUG) [15]. To facilitate standardization of newly generated compounds, this is also directly possible from MassSpecGym. Compounds whose SMILES representation failed standardization were discarded.

## 2.5  Data splitting

To obtain the training-validation-test split, we apply an MCES-based clustering technique. First, we compute the matrix of pairwise MCES distances between all molecules in the dataset with unique PubChem-standardized SMILES strings. Here, MCES distances are computed using the threshold of 10 with the `always_stronger_bound` option disabled. Then, we use this matrix to perform agglomerative clustering with the minimum distance as the cluster linkage criterion. Clusters are linked together only if the minimum distance is lower than 10. Specifically, we use the following initialization of the agglomerative clustering from the scikit-learn Python package [17]: `AgglomerativeClustering(metric=``precomputed'', linkage=``single'',`

Table 2: **Composition of the dataset split with respect to the number of distinct spectra, molecules, and MCES-based clusters**. The subtable below "All metadata available" describes the simulation challenge subset.

| | Num. spectra | Num. molecules | Num. clusters |
|---|---|---|---|
| Training | 194,119 (84%) | 25,046 (79%) | 3,061 (41%) |
| Validation | 19,429 (8%) | 3,386 (11%) | 2,221 (30%) |
| Testing | 17,556 (8%) | 3,170 (10%) | 2,202 (29%) |
| *All metadata available* | | | |
| Training | 101,573 (84%) | 13,543 (74%) | 2,628 (41%) |
| Validation | 9,975 (8%) | 2,445 (13%) | 1,917 (30%) |
| Testing | 10,159 (8%) | 2,417 (13%) | 1,907 (30%) |

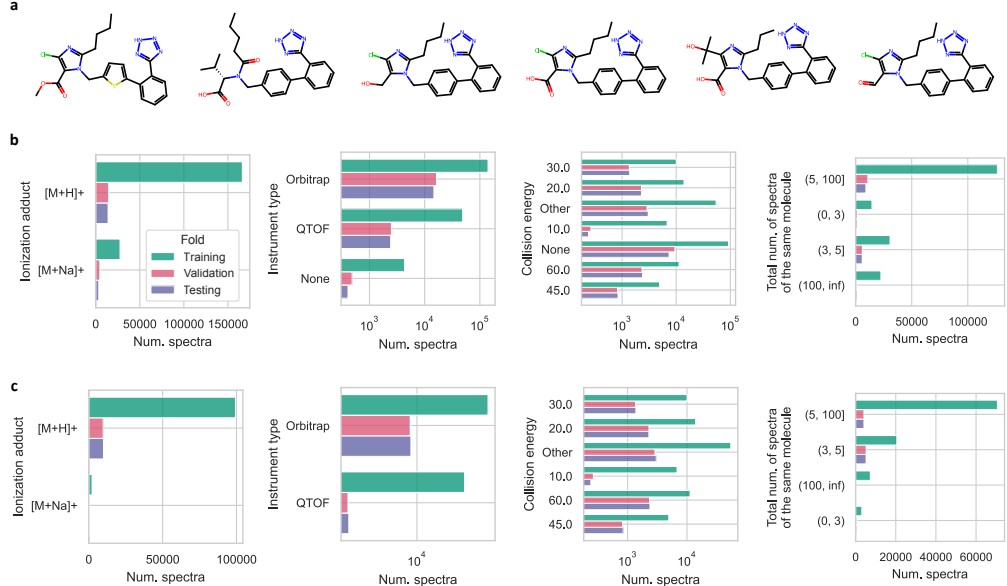

Figure 3: **MCES-based metadata-stratified data split results in a balanced composition of metadata across data folds. a,** Example of an MCES cluster of size six. **b,** Number of spectra in each fold in the dataset with respect to different metadata properties. **c,** Same as **b**, but for the subset of the dataset with no missing metadata. This subset is used for the spectrum simulation challenge.

`distance_threshold=10, n_clusters=None)`. Figure 3a shows an example of a cluster. We leverage the computed clusters as groups for the `StratifiedGroupKFold` splitting algorithm, assigning each molecule to one of the three folds. We use the concatenated string representations of ionization adducts, collision energies, instrument types, and the numbers of spectra per molecule as the stratification labels.

Since the majority (65%) of molecules formed a single cluster, the initial splitting resulted in a 74%-13%-13% composition of training-validation-test folds in terms of the number of dataset entries (i.e., spectra) and a 65%-18%-17% composition in terms of molecular structures. Such proportions are generally acceptable for the training-validation-test split. However, we found that the resultant split underrepresents molecules in the training dataset when considering a subset with all metadata available (55%-22%-22%), as required for the simulation challenge. To address this issue, we reassigned randomly sampled validation and testing clusters to the training fold to obtain a more balanced composition of the split with respect to molecular structures (74%-13%-13%). Table 2 shows the final proportions after the reassignment. Figure 3b and Figure 3c show the balanced proportions of metadata values across folds for the full dataset and the metadata-complete subset respectively. Figure 4 shows the balanced distribution of chemical classes across the entire dataset.

## 2.6 Auxiliary unlabeled datasets

Additionally, as a part of our MassSpecGym dataset, we provide curated unlabeled datasets of mass spectra and molecules. For the mass spectra, we provide the GeMS-A10 dataset, a deduplicated collection of 24 million high-quality mass spectra mined from the MassIVE GNPS repository [18]. As detailed in the main text, for the molecules, we provide three datasets of increasing size and decreasing relevance for mass spectrometry applications: a 1-million set [3] and a 4-million set [19] of biologically significant molecules, and all 118 million molecules (as of May 31, 2024) from PubChem [20].

We use the sets of molecules to construct retrieval candidates for the challenges of molecule retrieval and spectrum simulation. The procedure is outlined in Algorithm 1. The algorithm iteratively samples candidates from given molecular databases that are similar to the query molecule until the maximum number of candidates is reached. The similarity is measured either by similar mass (up

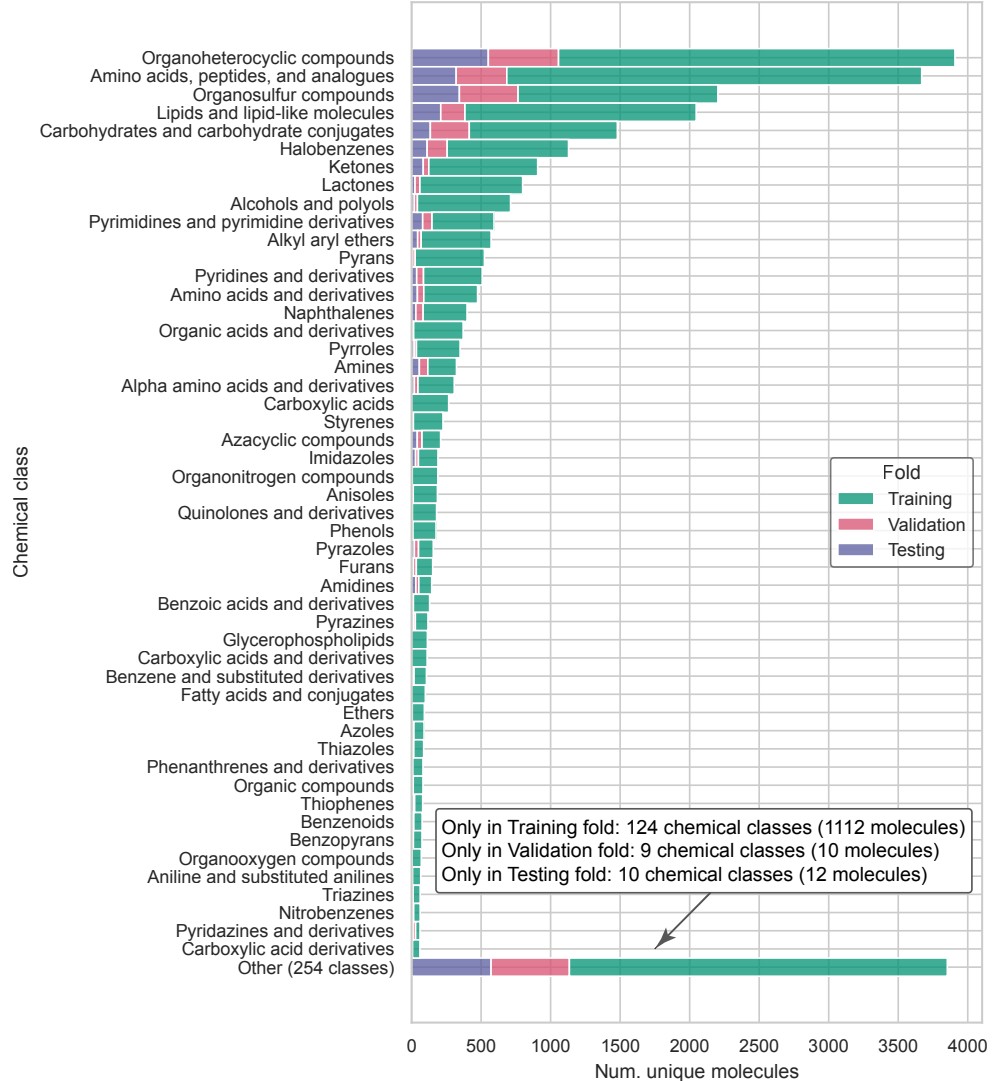

Figure 4: **MCES-based metadata-stratified data splitting results in a balanced distribution of chemical classes across data folds.** The figure presents a histogram of the 50 most common chemical classes according to ClassyFire [16], found in MassSpecGym, with a separate bar for all less common classes, labeled as "Other (254 classes)". The box with an arrow pointing to this bar shows the number of classes uniquely present in individual folds, along with the number of underlying molecules.

to experimental error) for the standard challenges, or identical formula for the bonus challenges. In practice, we use the three chemical databases mentioned above as $\mathcal{D}$ and the maximum number of candidates $|C| = 256$. The statistics and examples of retrieval candidates are visualized in Figure 5.

When developing new methods that leverage unlabeled data, we anticipate that users will rely solely on the following two datasets: the GeMS-A10 dataset of unlabeled MS/MS spectra and a refined subset of the unlabeled 4 million-molecule dataset. The molecular dataset has been refined by excluding any molecules with an MCES distance of less than two from any molecule in the test fold of MassSpecGym. This refinement is intended to reduce the potential for data leakage, particularly when used in the context of the *de novo* generation challenge.

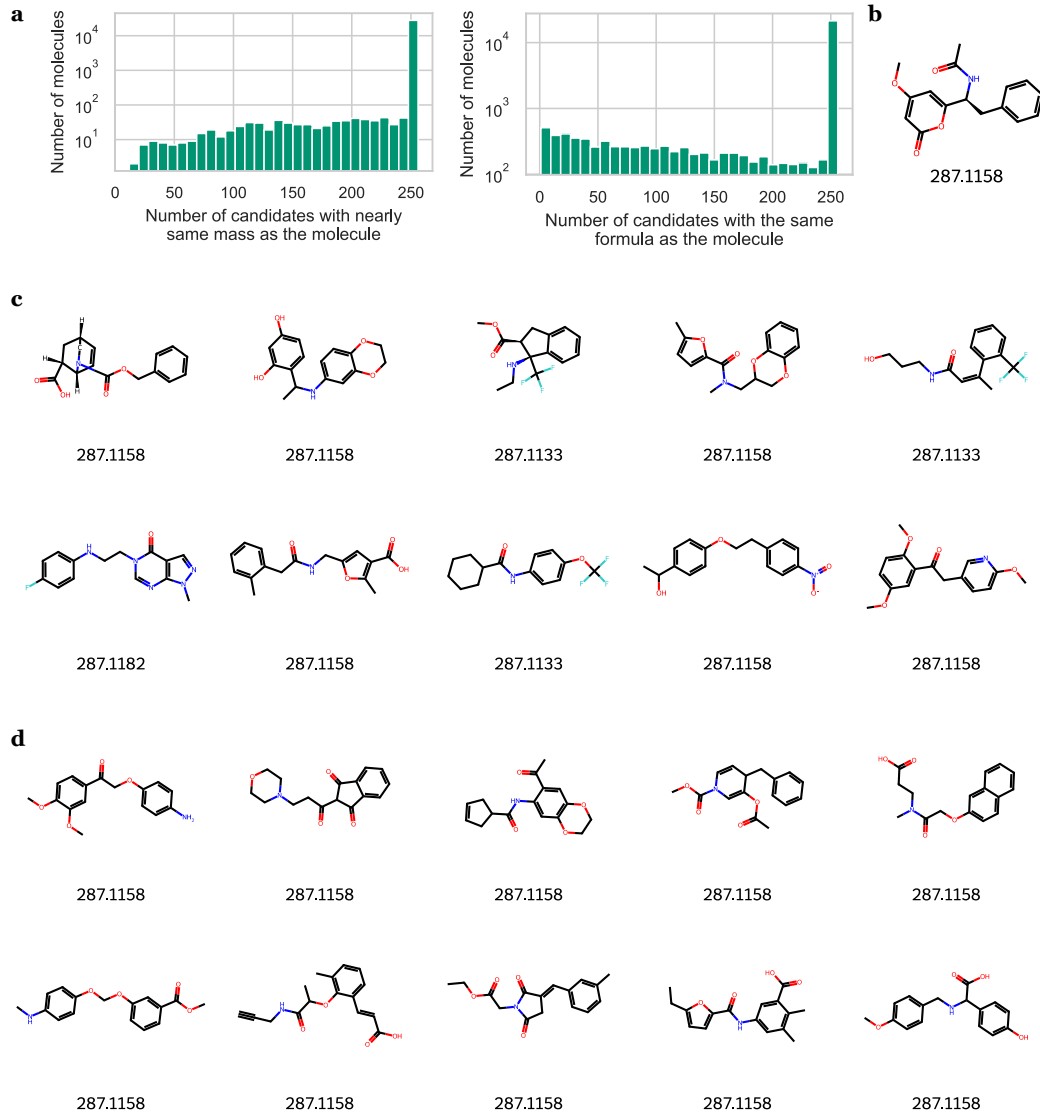

Figure 5: **Overview of molecule retrieval candidates. a,** The distributions of the number of retrieval candidates per molecule in the MassSpecGym dataset. The left histogram corresponds to candidates for the main molecule retrieval challenge, where the candidates are selected to have a mass similar to the query molecule, and the right histogram corresponds to the chemical formula bonus challenge, where candidates are restricted to having the same formula as the query molecule. Most of the molecules (87% and 66% respectively) have the predefined maximum number of 256 candidates. **b,** Example of a molecule having 256 both mass-based and formula-based candidates. **c,** Ten randomly sampled mass-based candidates for the molecule shown in figure **b**. **d,** Ten randomly sampled formula-based candidates for the molecule shown in figure **b**. The numbers below the molecules show their mass (in Daltons).

## 2.7 Dataset limitations

MassSpecGym aims to make machine learning applied to MS/MS spectra accessible to the machine learning community and rigorously standardized. To achieve this, we had to make certain simplifications that inherently limit MassSpecGym. For example, we focused exclusively on MS/MS spectra acquired in positive ionization mode, retained only spectra with the most common ionization adducts and with singly-charged ions, and sought to eliminate noisy spectra. However, this approach does not

**Algorithm 1** Construction of retrieval candidates

---

**Require:** Query molecule $q$, kind of candidates $kind \in \{\texttt{mass}, \texttt{formula}\}$, list of chemical databases with increasing significance $\mathcal{D}$ to sample candidates from, maximum number of candidates $N$
**Ensure:** Candidate set $\mathcal{C}$
    $\mathcal{C} \leftarrow \{q\}$
    **if** $|\mathcal{C}| = N$ **then**
        **return** $\mathcal{C}$
    **end if**
    **for** $D \in \mathcal{D}$ **do**
        **if** $kind = \texttt{mass}$ **then**
            $\epsilon \leftarrow mass(q) \times 10 \times 10^{-6}$                 ▷ 10 parts per million (ppm)
            $\mathcal{C} \leftarrow \mathcal{C} \cup \{c \in D \mid |mass(c) - mass(q)| < \epsilon \land inchi2d(c) \neq inchi2d(q)\}$
        **else if** $kind = \texttt{formula}$ **then**
            $\mathcal{C} \leftarrow \mathcal{C} \cup \{c \in D \mid formula(c) = formula(q) \land inchi2d(c) \neq inchi2d(q)\}$
        **end if**
    **end for**
    $\mathcal{C} \leftarrow \mathcal{C}[:N]$                             ▷ Get first $N$ elements added to $\mathcal{C}$
    **return** $\mathcal{C}$

---

fully reflect real-world scenarios where a significant portion of spectra may be instrument noise or acquired in different settings. Additionally, we were unable to adequately address certain types of noise, such as isotope signals in MS/MS spectra or chimeric spectra representing multiple molecules, due to the lack of appropriate metadata in public spectral libraries. Our dataset is also combined from highly heterogeneous and imbalanced spectral libraries, which may pose challenges when training machine learning models. Finally, our work is solely focused on MS/MS spectra, not considering other types of spectra, such as electron ionization (EI) or nuclear magnetic resonance (NMR) spectra.

## 3 Evaluation of baselines

In this section, we present a detailed description of the architectures, loss functions, and hyper-parameter configurations for each model. Unless stated otherwise, we train the models using all possible combinations of hyperparameters and select the best model for testing based on the minimum validation loss. To prevent overfitting, training is stopped if there is no improvement in validation loss over the last three validation epochs. Unless specified, we train the models on four AMD MI250X GPUs (8 PyTorch devices) using the distributed data parallel (DDP) mode.

### 3.1 De novo molecule generation

**Random chemical generation.** The goal of the challenge is to assess the capabilities of predictive models to extract information about molecules from their mass spectra. A potential failure mode for these models is to ignore the input mass spectra and generate molecules solely from prior chemical knowledge. To address this risk, we implement a baseline that generates molecular structures using only prior chemical knowledge.

In detail, the baseline consists of combinatorial and graph algorithms. It begins by identifying a chemical formula from the training data with the closest molecular weight. Then, the procedure iterates over plausible combinations of atom valence assignments that satisfy the selected chemical formula and can form a feasible combination of covalent and coordinate bonds in a molecule. Finally, through random graph traversal, we generate a connected molecular structure that respects the assigned atomic properties (valence and charge). Additionally, the baseline samples the edges in molecules based on the edge distribution observed in the training data structures.

**SMILES Transformer.** The SMILES Transformer model uses a standard encoder-decoder architecture [21] with post-norm and is trained using standard teacher forcing method. The input is given by 2D continuous tokens capturing m/z and intensity values of the spectrum peaks. Additionally, we add a token representing precursor m/z with a dummy intensity equal to 1.1. We linearly project the tokens to have dimensions corresponding to the hidden dimension of the model. The output SMILES

Table 3: **Hyperparameter grid explored for SMILES Transformer and SELFIES Transformer on the *de novo* generation challenge.** The optimal values, leading to the minimum validation loss, are highlighted in bold. For the bonus chemical formulae challenge, we use the same hyperparameters.

| Hyperparameter | Values (SMILES Transformer) | Values (SELFIES Transformer) |
|---|---|---|
| Learning rate | $\mathbf{3 \cdot 10^{-4}}, 1 \cdot 10^{-4}, 5 \cdot 10^{-5}$ | $\mathbf{3 \cdot 10^{-4}}, 1 \cdot 10^{-4}, 5 \cdot 10^{-5}$ |
| Batch size (per GPU) | 64, **128** | 64, **128** |
| $k$ predictions | **10** | **10** |
| Transformer hidden dimensionality | **256**, 512 | **256**, 512 |
| Number of attention heads | **4**, 8 | 4, **8** |
| Number of encoder layers | **3**, 6 | 3, **6** |
| Sampling temperature | 0.8, **1.0**, 1.2 | **1.0** |

Table 4: **Hyperparameter grid explored for Fingerprint FFN on the molecule retrieval challenge.** The optimal values, leading to the minimum validation loss, are highlighted in bold. For the bonus chemical formulae challenge, the optimal hyperparameters remain the same, except for an increased batch size of 64.

| Hyperparameter | Values |
|---|---|
| Learning rate | $3 \cdot 10^{-4}, \mathbf{1 \cdot 10^{-3}}$ |
| Batch size (per GPU) | **16**, 64, 128 |
| Hidden dimensionality | **128**, 1024 |
| Number of layers | 3, **7** |
| Dropout | 0.0, **0.1** |

is postprocessed using a byte-pair encoder [22] to increase the chance of generating valid molecules. The byte-pair encoder is trained on the 4M set of molecules discussed in Section 2.6. We use greedy decoding by sampling each token from the predicted distribution over the softmax vocabulary with a given temperature to predict $k$ samples for each input. For early stopping, we monitor the validation loss once per epoch. The training of the best model took 24 epochs (1 hour) in total. Hyperparameters used in the model are listed in Table 3.

For the bonus chemical formulae challenge, we slightly modify the model to condition it on the chemical formula of a target molecule. Specifically, we feed the formula into the encoder alongside the MS/MS spectrum using an additional feed-forward network. This network maps the vectorized chemical formula (where each element represents the count of a specific chemical element) to the transformer's hidden dimensionality. The output from this network is then added to the embedding of each token before it enters the transformer layers. We use a single hidden layer with a dimensionality matching the number of considered chemical elements, which is 118.

**SELFIES Transformer.** The SELFIES Transformer model follows the same implementation as the SMILES Transformer, with the exception of using SELFIES molecular string representations [23] instead of SMILES. Additionally, we do not use byte-pair encoding for this model, as the SELFIES grammar is specifically designed for generating molecular graphs that are both syntactically and semantically valid.

The optimal hyperparameters for SELFIES Transformer are provided in Table 3, and the training of the best model took 25 epochs (1 hour) in total.

## 3.2 Molecule retrieval

**Fingerprint FFN.** The Fingerprint FFN baseline employs a simple feedforward neural network to predict Morgan fingerprints of molecules from binned MS/MS spectra. Specifically, we convert an input spectrum using binning with a maximum m/z value of 1005 and a bin width of 1 m/z. The network outputs a 4096-dimensional vector, trained to approximate the true Morgan fingerprint (with a radius of 2) of the underlying molecule, using cosine similarity as the loss function. Once the fingerprint is predicted for a spectrum, we sort the corresponding candidate list by cosine similarity

to obtain the final top-$k$ predictions. For the chemical formulae bonus challenge, we use a different set of candidates for each spectrum, pruned to include only molecules with the same formula.

We experiment with various standard feedforward neural network hyperparameters. The grid of values explored in this work is provided in Table 4. We monitor the validation loss twice per epoch. The training of the best model took 4 epochs (6.5 hours) for the standard challenge and 5 epochs (7 hours) for the bonus challenge.

Table 5: **Hyperparameter grid explored for DeepSets on the molecule retrieval challenge.** The optimal values, leading to the minimum validation loss, are highlighted in bold. For the bonus chemical formulae challenge, the optimal hyperparameters for the architecture remain the same while the training parameters are a learning of $1 \cdot 10^{-3}$ and a batch size of 128.

| Hyperparameter | Values |
|---|:---:|
| Learning rate | $\mathbf{3 \cdot 10^{-4}}, 1 \cdot 10^{-3}$ |
| Batch size (per GPU) | $\mathbf{16}, 64, 128$ |
| Hidden dimensionality | $\mathbf{128}, 1024$ |
| Number of layers (per MLP) | $2, \mathbf{4}$ |
| Dropout | $0.0, \mathbf{0.1}$ |

**DeepSets.** Since the binning of spectra has the drawback of producing sparse input vectors and ambiguously rounding signals at the edges of m/z bins, we implement a DeepSets baseline [24] that treats MS/MS spectra as sets of two-dimensional elements: m/z and intensity values of individual signals. The output of DeepSets is a 4096-dimensional Morgan fingerprint, similar to Fingerprint FFN. The entire model is represented as:

$$\hat{\mathbf{y}} = \rho \left( \sum_{i=1}^{n} \phi(\mathbf{m}_i \| \mathbf{i}_i) \right), \tag{4}$$

where $\hat{\mathbf{y}}$ is the predicted fingerprint, $\phi : \mathbb{R}^2 \to \mathbb{R}^d$ and $\rho : \mathbb{R}^d \to \mathbb{R}^{4096}$ are feed-forward neural networks, $d$ is the hidden dimensionality, $n$ is the number of signals in the input MS/MS spectrum, $\mathbf{m}_i$ and $\mathbf{i}_i$ are m/z and intensity values of the $i$-th signal respectively. $\|$ denotes concatenation.

Similar to the Fingerprint FFN baseline, we experiment with different hyperparameter setups (Table 5), monitoring validation performance twice per epoch for early stopping. The training of the best model took 3 epochs (10.5 hours) for the standard challenge and 1 epoch (5 hours) for the bonus challenge.

**DeepSets + Fourier features.** To enhance the representation of m/z values in the DeepSets architecture, we process them using the Fourier features technique [25]. Specifically, we pre-process each m/z value with a set of sine and cosine functions with 6,000 pre-defined wavelengths, following [18]. These wavelengths decompose each m/z value into 6,000 input features, capturing its both integer and decimal components with greater sensitivity than the single value. This allows the model to be, for example, more sensitive to small differences in m/z values, which are particularly relevant when working with high-accuracy mass spectrometers. The overall DeepSets architecture is then modified as follows:

$$\hat{\mathbf{y}} = \rho \left( \sum_{i=1}^{n} \phi \left( \mathbf{W}_1 \text{FOURIER}(\mathbf{m}_i) + \mathbf{b}_1 \| \mathbf{W}_2 \mathbf{i}_i + \mathbf{b}_2 \right) \right), \tag{5}$$

where $\text{FOURIER} : \mathbb{R} \to \mathbb{R}^{6000}$ represents the Fourier features, and $\mathbf{W}_1 \in \mathbb{R}^{d_1 \times 6000}$, $\mathbf{W}_2 \in \mathbb{R}^{d_2 \times 1}$, $\mathbf{b}_1 \in \mathbb{R}^{d_1}$, and $\mathbf{b}_2 \in \mathbb{R}^{d_2}$ are learnable parameters. Here, $d_1 = \lceil 0.8d \rceil$, $d_2 = d - d_1$, and $\phi$ takes $d$ inputs instead of 2.

For the DeepSets + Fourier features model, we use the final hyperparameters from the DeepSets model.

Table 6: **Hyperparameter values explored for MIST for the molecule retrieval challenge.** No grid search was performed due to the computational cost of training MIST. Instead, 5 combinations of hyperparameter values were selected for training. The optimal values, leading to the minimum validation loss, are highlighted in bold.

| Hyperparameter | Values |
|---|---|
| Learning rate | 0.0001, **0.0003** |
| Scheduler | True, **False** |
| Weight decay | 1e-6, **1e-7** |
| Learning rate decay | **0.9**, 0.995 |
| Dropout | 0, **0.3** |
| Hidden size | **512** |
| Number of layers | **3**, 5 |
| Number of unfolding layers | **5** |
| Unfolding loss weight | **0.1** |
| Magma loss weight | 2, **4** |
| Probability noised spectrum | **0.5** |
| Probability remove peak | **0.5** |
| Probability scale peak | **0.1** |
| Batch size | **128** |

**MIST.** The Metabolite Inference with Spectrum Transformers (MIST) [26] tool is employed as the state-of-the-art baseline for the molecule retrieval challenge. MIST utilizes a deep learning approach to annotate mass spectra with chemical structures. This tool integrates domain knowledge into its architecture by representing peaks with their associated chemical formulae, which are then processed by the specialized "chemical formula transformer". Furthermore, MIST predicts low-resolution fingerprints, which are subsequently refined to achieve full resolution. By predicting these fingerprints and matching them against a database of candidate structures, MIST effectively annotates MS/MS spectra.

Before training, the dataset is processed to meet the requirements of MIST, including converting spectra to `.ms` files, subformula labeling, and MAGMa [27] substructure annotation. MIST v2.0.0 is trained using the hyperparameters specified in Table 6. No simulated spectra are used during training. All relevant code can be found on `https://github.com/Janne98/mist/tree/mass-spec-gym`.

We train MIST on a node equipped with two AMD Epyc 7452 Zen2 CPUs (totaling 64 cores) and a single NVIDIA Ampere A100 GPU. The training process utilizes early stopping with a patience of 20 epochs. The validation loss is computed after each epoch. Convergence is achieved after 32 epochs, which takes approximately 5 hours. The final model weights and the processed dataset can be found on `https://zenodo.org/records/11580401`

### 3.3 Spectrum simulation

**FFN Fingerprint.** The FFN Fingerprint model is based on an implementation of the NEIMS model [28] from [29]. Let $G$ be the input molecule, and $p_{FP}(G) \in \mathbb{R}^n$ be its molecular fingerprint featurization. In practice we use a combination of three molecular fingerprints: the ECFP4 (Morgan) fingerprint [1], the rdkit fingerprint [30], and the MACCS fingerprint [31]. Let $z \in \mathbb{R}^m$ be a vector representation of metadata for the spectrum prediction (collision energy, instrument type, and precursor adduct). Let $f_{FP} : \mathbb{R}^n \times \mathbb{R}^m \to \mathbb{R}^d$ be a neural network that maps from the fingerprint to the binned spectrum, where $d$ is the number of bins. The architecture of $f_{FP}$ is an MLP with skip-connections. To predict the output spectrum, our model uses gated bidirectional spectrum prediction (refer to [28] for full details). The loss function is cosine distance, although the intensities of the binned target spectrum are subjected to a square-root transform before distance calculation.

Key hyperparameters for the FP model were optimized using a random sweep with a budget of 50 samples. The results of the sweep are summarized in Table 7. Performance was measured on the simulation split validation set (roughly 10% of all simulation spectra), and the configuration with the highest score was selected for testing. All models were trained for 100 epochs. The learning rate followed a linear decay schedule following a warmup of 1000 steps. The "Precursor m/z offset"

Table 7: **FFN Fingerprint model hyperparameter sweep configuration and optimal values for the spectrum simulation challenge.**

| Hyperparameter | Possible Values | Optimal Value |
|---|---|---|
| Learning rate | [1e-4, 1e-3] | 1.563e-4 |
| Weight decay | $\{0, 1e\text{-}7, 1e\text{-}6\}$ | 1e-7 |
| Learning rate decay | [0.7, 1.0] | 0.9866 |
| Batch size | $\{32, 64, 128, 256\}$ | 256 |
| MLP Dropout | $\{0, 0.1, 0.2, 0.3, 0.4, 0.5\}$ | 0 |
| MLP Hidden size | $\{256, 512, 1024\}$ | 1024 |
| MLP Number of layers | $\{1, 2, 3, 4, 5\}$ | 5 |
| Precursor m/z offset | $\{5, 50, 500\}$ | 5 |

Table 8: **Molecular graph node and edge features for the GNN model and the FraGNNet model.**

| Feature | Possible Values |
|---|---|
| Atom Type (Element) | {C, O, N, P, S, F, Cl, Br, I, Se, Si} |
| Atom Degree | $\{0, \ldots, 10\}$ |
| Atom Orbital Hybridization | {SP, SP2, SP3, SP3D, SP3D2} |
| Atom Formal Charge | $\{-2, \ldots, +2\}$ |
| Atom Radical State | $\{0, \ldots, 4\}$ |
| Atom Ring Membership | {True, False} |
| Atom Aromatic | {True, False} |
| Atom Mass | $\mathbb{R}^+$ |
| Atom Chirality | {Unspecified, Tetrahedral CW, Tetrahedral CCW} |
| Bond Degree | {Single, Double, Triple, Aromatic} |

parameter refers to the offset used for bidirectional prediction, corresponding to the $\tau$ parameter in Equations 4 and 5 of [28].

**GNN Model.** The GNN model is based on an implementation from [29]. Given the molecular graph $G = (V, E)$, a set of node and edge features are $X_V = \{X_v\}_{v \in V}$ and $X_E = \{X_e\}_{e \in E}$ are extracted (see Table 8). These features are passed to a GNN $g_{\text{GNN}}(G, X_V, X_E)$ which outputs a set of node embeddings $H_V = \{h_v \in \mathbb{R}^l\}_{v \in V}$. The GNN architecture is Graph Isomorphism Network with Edge Features [32], as implemented in the Pytorch Geometric library [33]. Average pooling is used to produce a graph-level embedding $h_G \in \mathbb{R}^l$ from the individual atom-level embeddings $H_V$. The molecule embedding is then concatenated with the metadata vector $z \in \mathbb{R}^m$ and passed to another neural network $f_{\text{GNN}}(h_G, z)$ that makes a spectrum prediction. The architecture of $f_{\text{GNN}}$ is identical to that of $f_{\text{FP}}$, albeit with different hyperparameter configurations and input dimensions. The loss function and intensity transformations are consistent with the FFN Fingerprint model.

As was the case with the FFN Fingerprint model, a hyperparameter sweep with a budget of 50 samples was used to select hyperparameters for the GNN model. The results of this sweep are summarized in Table 9.

**FraGNNet Model.** The FraGNNet model [29] simulates a spectrum by modelling a distribution over fragments of the input molecule $G$ and then mapping this distribution to a mass spectrum; for full details, refer to [29]. A recursive fragmentation algorithm $p_{\text{FRAG}}(G)$ is used to pre-process $G$ into a fragmentation DAG $G_D = (V_D, E_D)$ that represents a hierarchy of plausible molecular fragments. Each node $s$ of $V_D$ represents a fragment (connected subgraph) of the original molecular graph $G$. A GNN $g^1_{\text{FRAG}}(G, X_V, X_E)$, with identical architecture to $g_{\text{GNN}}$ (excluding hyperparameter choices), is used to generate atom embeddings $H_V$. The information from the DAG $G_D$ and the atom embeddings $H_V$ is jointly processed by a second GNN $g^2_{\text{FRAG}}(G_D, H_V)$, which produces embeddings $H_S = \{h_s \in \mathbb{R}^k\}_{s \in G_V}$. An MLP $f_{\text{FRAG}} : \mathbb{R}^k \to \mathbb{R}$ is applied to each fragment embedding to predict a distribution over fragments. The mass spectrum (which is a distribution over masses) can be inferred from the fragment distribution by aggregating probabilities for fragments that share the same chemical formula and using the masses of these formulae (which are known) to map the probabilities

Table 9: **GNN hyperparameter sweep configuration and optimal values for the spectrum simulation challenge.**

| Hyperparameter | Possible Values | Optimal Value |
|---|---|---|
| Learning rate | $[1\text{e-}4, 1\text{e-}3]$ | 3.755e-4 |
| Weight decay | $\{0, 1\text{e-}7, 1\text{e-}6\}$ | 1e-6 |
| Learning rate decay | $[0.7, 1.0]$ | 0.8523 |
| Batch size | $\{32, 64, 128, 256\}$ | 128 |
| MLP Dropout | $\{0, 0.1, 0.2, 0.3\}$ | 0 |
| MLP Hidden size | $\{256, 512, 1024\}$ | 512 |
| MLP Number of layers | $\{2, 3, 4, 5\}$ | 4 |
| Precursor m/z offset | $\{5, 50, 500\}$ | 5 |
| GNN Normalization | $\{none, batch, layer, graph\}$ | batch |
| GNN Dropout | $\{0, 0.1, 0.2, 0.3\}$ | 0.2 |
| GNN Hidden size | $\{64, 128, 256, 512\}$ | 256 |
| GNN Number of layers | $\{2, 3, 4, 5, 6\}$ | 4 |

Table 10: **FraGNNet hyperparameter values for the spectrum simulation challenge.**

| Hyperparameter | Value |
|---|---|
| Learning rate | 3.033e-4 |
| Weight decay | 0.01 |
| Batch size | 128 |
| MLP Dropout | 0.2 |
| MLP Hidden size | 64 |
| MLP Number of layers | 2 |
| Mol GNN Normalization | batch |
| Mol GNN Dropout | 0.2 |
| Mol GNN Hidden size | 256 |
| Mol GNN Number of layers | 4 |
| Frag GNN Normalization | graph |
| Frag GNN Dropout | 0.1 |
| Frag GNN Hidden size | 256 |
| Frag GNN Number of layers | 4 |

to a mass distribution. The model is trained by minimizing the cross-entropy between the predicted distribution and the ground-truth spectrum. Although FraGNNet can model peak locations with extremely high precision, during evaluation we bin the predicted spectrum at 0.01 Da to allow for fair comparison with the other baseline models.

The hyperparameters for the FraGNNet model were not optimized using a sweep, since the initial values significantly outperformed the other two simulation models. The key hyperparameter selections are summarized in Table 10. Unlike the other baselines, a learning rate schedule was not applied during training.

### 3.3.1 Jensen-Shannon Similarity Metric

The Jensen-Shannon similarity metric is an alternate way of representing Jensen-Shannon divergence between the true mass spectrum $x$ and the predicted mass spectrum $\hat{x}$, both of which can be interpreted as discrete probability distributions. It is defined using the following equation:

$$JSS(x, \hat{x}) = 1 - \frac{JSD(x, \hat{x})}{\log(2)} \tag{6}$$

Note that $JSD$ denotes the Jensen-Shannon divergence calculated with the natural logarithm $\log$. $JSD$ ranges from $\log(2)$ (when $x$ and $\hat{x}$ share no support) to 0 (when $x = \hat{x}$); $JSS$ ranges from 0 to 1.

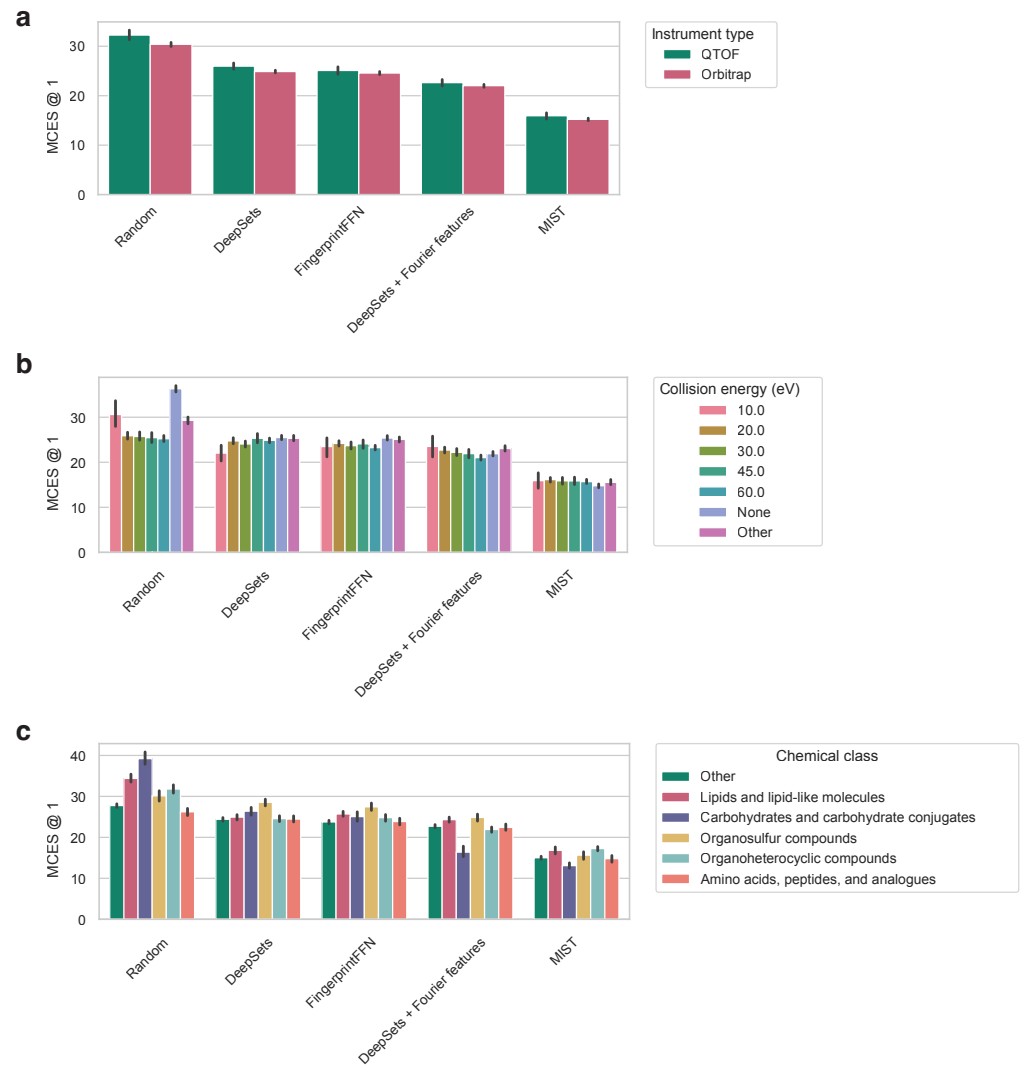

Figure 6: **Test performance of the MIST model on the molecule retrieval challenge, stratified by various metadata. a,** Performance stratified by instrument types. **b,** Performance stratified by collision energies. **c,** Performance stratified by the top 5 most common chemical classes. Error bars represent 99.9% confidence intervals, calculated through bootstrapping with 20,000 resamples.

## 3.4 Extended results

In addition to the overall metrics averaged across the entire test set, as presented in the main text, we evaluate the MCES @ 1 metric on the molecule retrieval challenge across different metadata subsets. Specifically, we stratify the results of all baselines by instrument types, collision energies, and chemical classes (Figure 6). Overall, all methods perform better on MS/MS spectra from Orbitrap instruments, which generally produce higher-quality spectra. Regarding collision energies and chemical classes, machine learning methods perform relatively consistently across various subsets, with the exception of DeepSets + Fourier features, which excels on carbohydrates and carbohydrate conjugates. Interestingly, this chemical class exhibits the highest variability in performance across methods, while classes such as organosulfur compounds result in more limited variability.

# 4 Background on mass spectrometry

This section introduces mass spectrometry to the broader machine learning community. Based on [34], we first define the concept of molecular structures (Section 4.1) and then outline the basic principles of tandem mass spectrometry (Section 4.2).

## 4.1 Small molecules

An **atom** is the fundamental building block of matter. Each atom is formed of **protons** and **neutrons** comprising a nucleus, as well as **electrons** orbiting around the nucleus. The number of protons defines a **chemical element** of an atom. For example, H (hydrogen) atom has 1 proton, C (carbon) atom has 6 protons, Br (bromine) atom has 35 protons, and so on. A **molecule** is a compound made up of two or more atoms that are chemically **bonded** together by sharing electrons. The more pairs of electrons they share the stronger the bond between atoms. **Molecular structure** is typically understood as a labeled undirected graph, where nodes represent atoms and are labeled by the corresponding chemical elements along with their spatial coordinates. Edges represent bonds and are labeled by the number of shared electron pairs.

3D Ball-and-stick representation

2D Skeletal structure

Figure 7: **Example molecular structure of *firefly luciferin* – a compound responsible for the characteristic yellow light emission from many firefly species.**

**Molecular representations.** Molecules containing carbon-hydrogen bonds are named **organic compounds**. Since they constitute the majority of known chemicals, it is convenient to represent them using simplified planar graph representations - **skeletal structures**. The example is shown in fig. 7. In a skeletal structure, nodes are implicitly associated with carbon atoms, and hydrogens adjacent to carbons are omitted. Noteworthy, this adaptation does not affect the expressivity of the representation because hydrogens can be unambiguously filled based on the bonding capacity of each element given by its composition. A spatial arrangement of atoms is encoded in special **stereochemical** types of bonds visualized as either dashed or solid triangles representing two opposite directions orthogonal to a molecular plane. Molecules differing only in stereochemical bonds are referred to as **stereoisomers**.

In order to simplify computer storage and processing, molecular structures are commonly encoded as strings. The three most widely used variants are **SMILES** (Simplified molecular-input line-entry system), **InChI** (International Chemical Identifier), and **InChIKey**. Although both SMILES and InChI serve the purpose of uniquely identifying a molecule as a sequence of characters, SMILES are more human-readable and simpler but do not undergo a unified standard. In contrast, InChI strings have more complex yet standardized grammar. To give an example, SMILES string of *firefly luciferin*, depicted in Figure 7, is C1[C@@H](N=C(S1)C2=NC3=C(S2)C=C(C=C3)O)C(=O)O, while its InChI string is InChI=1S/C11H8N2O3S2/c14-5-1-2-6-8(3-5)18-10(12-6)9-13-7(4-17-9)11(15)16/h1-3,7,14H,4H2,(H,15,16)/t7-/m1/s1. InChIKey representations are fixed-size hashes (e.g., IWJYWBVPCGUPLO-BFUDMSGGSA-N) derived from InChI strings, which are convenient, for instance, to perform searches of molecules in large databases. First 14 characters of the InChI key encode the connectivity information and are often referred to as **2D InChI keys**. Another compact coarse-grained representation of a molecule is its **chemical formula**, which represents the histogram of chemical elements within a molecule. Accordingly, the chemical formula of *firefly luciferin* is $C_{11}H_8N_2O_3S_2$.

The comparison of molecular structures is commonly conducted by utilizing their **fingerprints**, which are fixed-size binary vectors. A basic example of a molecular fingerprint is the Molecular ACCess System (MACCS) [35], where each of its 166 bits represents the presence or absence of a specific predefined substructure. The most widely-adopted family of fingerprints is Morgan fingerprints [1], which are fixed-size hashes encoding the local neighborhoods of molecular atoms. To compare two molecules, the most common approach is to generate the corresponding fingerprints and compute their Tanimoto similarity. The Tanimoto similarity is defined as a ratio of the number of shared positive bits to the total number of unique positive bits present in both fingerprints.

**Molecular properties.** **Molecular mass** is a central notion for mass spectrometry. It is characterized as a sum of **atomic masses** constituting the molecule and is usually measured in **Da** (Daltons). Single Dalton is defined as $\frac{1}{12}$ of the mass of $^{12}$C (carbon atom containing 6 protons and 6 neutrons). Importantly, a molecule is roughly termed as a "**small molecule**" if its mass is less than 1000 Da. As a consequence of the definition of a Dalton and the significantly smaller mass of electrons compared to protons and neutrons, one nuclear particle has a mass approximately equal to 1 Da. Specifically, a proton has a mass of approximately 1.007 Da, a neutron has a mass of approximately 1.009 Da, and an electron has a mass of approximately 0.0005 Da[1]. While the number of protons in the atom of a chemical element is given by the definition, the notation $^{12}$C is used to explicitly express the number of neutrons and protons.

Atoms having the same number of protons but different numbers of neutrons are referred to as **isotopes** of a chemical element. For instance, Cl (chlorine) element has two stable[2] isotopes: $^{35}$Cl having a mass of 34.96885269(4) Da and $^{37}$Cl having a mass of 36.96590258(6) Da. Naturally, $^{35}$Cl occurs in roughly 76% of cases and $^{37}$Cl occurs in remaining 24% of cases. Another element S (sulfur) has four stable isotopes: $^{32}$S, $^{33}$S, $^{34}$S, and $^{36}$S with natural abundances 94.99%, 0.75%, 4.25%, and 0.01% respectively. In contrast, F has only a single stable isotope $^{19}$F. The mass of the most abundant isotope is often termed as a **monoisotopic mass** of an element.

Another molecular property essential for the domain of mass spectrometry is a **molecular charge**. A molecule is defined as **negatively charged**, **neutral**, or **positively charged** if its atoms in total have more, equal, or fewer electrons than protons respectively. A charged molecule is termed **ion** and its charge is often expressed as an integer indicating the difference between the number of protons and electrons. The sign of such an integer can be often found in different notations and depictions of a molecule. For example, a positively-charged ion of a molecule M can be denoted as M+. **Ionization adducts** are formed when a molecule binds with an ion. Common examples include the [M+H]+ adduct, where a molecule M gains a proton (H+), and the [M+Na]+ adduct, where a molecule M gains a sodium ion (Na+).

## 4.2 Tandem mass spectrometry

The identification of molecules present in a sample is a fundamental task in various fields of biology and environmental science. To achieve this, Liquid Chromatography Tandem Mass Spectrometry (LC-MS/MS) is the most widely employed technique. This method constitutes an intricate pipeline composed of liquid chromatography and several stages of mass spectrometry (i.e., tandem mass spectrometry) to separate, elucidate, and quantify compounds in complex mixtures. Nevertheless, interpreting the output data from LC-MS/MS presents a significant challenge, as information about the molecules is only available in terms of their masses or the masses of their fragments. In the following section, we introduce tandem mass spectrometry. However, we do not discuss liquid chromatography, as it is outside the scope of this work.

**Acquisition of mass spectra.** Mass spectrometry (MS) plays a critical role in the workflow of LC-MS/MS, enabling the determination of the molecular mass of a compound. The fundamental principle of MS involves ionizing a sample to create charged molecules or fragments that are subsequently separated based on their mass-to-charge ratio (**m/z**) and detected using a mass analyzer. It is important to note that MS instruments are only capable of measuring m/z values and not the mass or charge[3] of the ions individually. The resulting **mass spectrum** is a collection of two-dimensional points, with the first dimension representing m/z values and the second dimension corresponding to their respective abundances (i.e. the **intensities** of the detected signals).

The ionization process in MS can occur through a variety of methods, including electron impact ionization (EI), electrospray ionization (ESI), and matrix-assisted laser desorption ionization (MALDI). Regardless of the method used, the result is the formation of ions with different m/z ratios, which are then separated by the mass analyzer. ESI is the most convenient and common method to couple with LC. During the electrospray ionization, the liquid sample is sprayed through a small needle

---

[1]Although the mass of an atom might be expected to equal the sum of the masses of its particles, it is always slightly less (with the exception of the hydrogen atom). This phenomenon is caused by the nuclear binding energy and is termed the mass defect of the nucleus.

[2]Isotope is stable if it does not decay into other elements on geologic timescales.

[3]Although, advanced mass spectrometry instruments report charges.

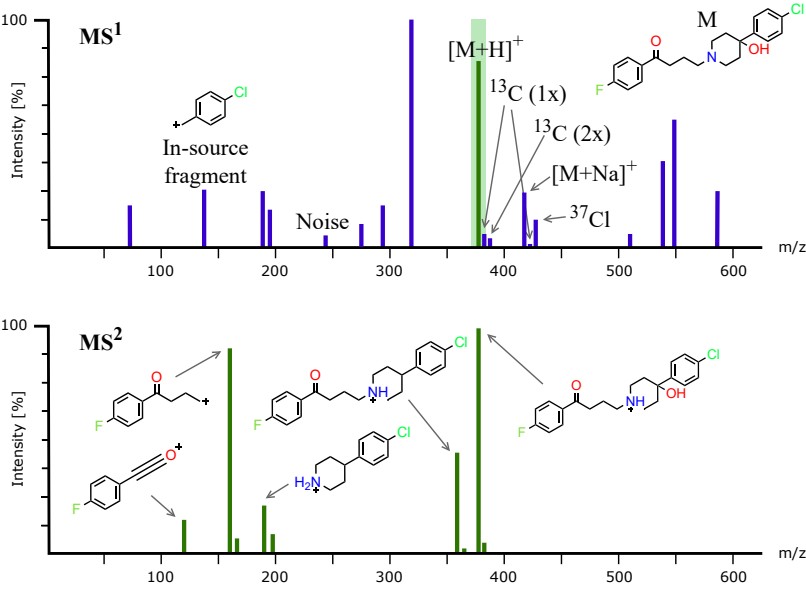

Figure 8: **Examples of MS$^1$ and MS/MS (i.e., MS$^2$) spectra. Top,** An example of an MS$^1$ spectrum, featuring haloperidol (designated as "M"), with a mass of 375.14 Da. Several peaks are labeled, while others may correspond to different compounds. Isotopes are denoted by $^{13}$C (1x) and $^{13}$C (2x), and $^{37}$Cl, indicating ions ([M+H]$^+$ or [M+Na]$^+$) containing corresponding isotopes. Notice, that the spectrum is simplified for visualization purposes, and one can frequently observe more intricate isotopic patterns and adduct species. **Bottom,** MS/MS (i.e., MS$^2$) spectrum acquired by fragmenting protonated haloperidol from MS$^1$ belonging to the isolation window highlighted in green.

that has a high voltage applied to it. The high voltage causes the liquid to form tiny droplets, and as these droplets move through the air, they pick up an electrical charge. These droplets will either be positively or negatively charged depending on the polarity of the applied voltage. Further, the charged droplets continue to break apart and re-form until they eventually become individual ions. During the ionization process, ions form **adducts**, which are clusters of molecules that stick together due to electrostatic interactions. For example, a molecule in the sample may pick up a positively charged droplet ion such as a proton, Na (sodium), or K (potassium). As a consequence, the resulting ion measured in a mass spectrum has a higher mass-to-charge ratio than the original molecule. Such adduct species of molecule M are then denoted as [M+H]+, [M+Na]+, or [M+K]+ respectively.

It is important to note that the understanding of the sample introduced to MS system should not be limited to a set of mutually exclusive molecules. Instead, it should be understood as a complex mixture of compounds with millions of duplicates for each molecular structure. For the sake of simplicity, we often refer to a "molecule" as a group of identical compounds in the state prior to the ionization. In fact, within a single mass spectrometry experiment one can frequently observe m/z values that correspond to various adduct species of "the same molecule", as well as differing isotopic compositions of "the same molecule" (Figure 8).

After ionization, the mass analyzer separates the ions based on their m/z ratios. There are various types of mass analyzers, such as time-of-flight (QTOF), quadrupole, and Orbitrap, each with unique strengths and weaknesses. However, they all operate on the principle of using a combination of electromagnetic fields to isolate ions. Finally, the ion detector, typically integrated into the mass analyzer, measures the m/z value and intensity of the signal. While we will not delve into the technicalities, it's worth noting certain peculiarities of the measurement that are significant for subsequent MS data analysis.

**Measurment of tandem mass spectra.** Tandem mass spectrometry (MS/MS or MS$^2$) is a powerful technique enabling the detailed analysis of molecular structures by breaking down molecules into

smaller fragments and analyzing their mass-to-charge ratios (m/z). It involves using two or more subsequent mass spectrometry experiments. The first stage of mass spectrometry ($\mathbf{MS^1}$) is out of the scope of this work; therefore, we do not discuss it here. Nevertheless, Figure 8 shows the main concept. In the second stage (**MS/MS** or $\mathbf{MS^2}$), the instrument selects specific ions of interest termed **precursor ions**. This is realized by defining an **isolation window** of specific width that slides across the m/z range to select ions of interest. The selected m/z values can be either arbitrary (data-independent acquisition; DIA) or pre-defined beforehand (data-dependent acquisition; DDA).

Once the precursor ion is identified, it is then subjected to fragmentation. The most prominent technique to fragment the ion is collision-induced dissociation (CID). It works by accelerating the molecule towards a gas, causing it to collide with the gas molecules and recursively break into smaller substructures. The more **collision energy** is provided, the more molecular fragments are obtained as a result. The second mass spectrometer is then used to measure the **MS/MS spectrum** (also referred to as $MS^2$ spectrum or fragmentation spectrum), where peaks represent m/z ratios of individual fragments. Additionally, the process of selecting and fragmenting ions can be recursively repeated up to the $MS^n$ level for any reasonable positive integer $n$ retaining fragments.

Because the instrument can only detect ionized fragments, only a portion of substructures are recorded. For instance, a singly-charged ion broken into two fragments will form one ion and one neutral molecule (**neutral loss**) depending on the current location of the charge within the molecule. However, since "precursor ion" is in fact a group of identical molecules and the charge site is not deterministic with respect to molecule, $MS^2$ spectrum often contains peaks for both parts with intensities reflecting their probability distribution. In particular, fragmentation spectrum often contains **precursor peak** corresponding to the whole non-fragmented precursor ion.

It is important to note that the graph-theoretical abstraction of fragmentation as a consequent removal of bonds is often, but not always, correct. For example, during CID fragmentation, a molecule can undergo rearrangement or transfer reactions leading to graph deformations, such as the formation of new rings [36, 37].

**MS/MS spectra and their properties.**   The detector records the entire ion signal as a function of time, resulting in a mass spectrum that exhibits a continuous signal proportional to the intensity of ions relative to their mass-to-charge (m/z) ratio. This type of spectrum is known as a **profile** mass spectrum. In contrast, the instrument often produces **centroid** spectra, which undergo pre-processing via an algorithm that extracts peak information from the profile mass spectrum. The resulting signals are reported at specific m/z values and are termed **peaks** or **signals**. Also, it is a common practice to pre-process intensities such that they are represented as fractions of the maximum spectrum intensity (corresponding to the **base peak**), and are referred to as **relative intensities**.

The accuracy of the measured m/z ratios is profoundly reliant on the instrument's quality and its constituents. The two most crucial indicators of quality are the **resolution** and **accuracy** of the instrument. Resolution denotes the ability to differentiate ions with nearly identical masses, whereas accuracy indicates how close the measured m/z value is to the actual ground-truth value. Since modern instruments generally possess high separation capabilities, the resolution is often not a problem for downstream data analysis. Nonetheless, accuracy remains a pivotal concept.

Instrument vendors typically provide the accuracy of individual instruments in **ppm** (parts per million), which means that accuracy is inversely proportional to the measured mass. Specifically, a ppm of 5 would indicate that for the ground-truth m/z $m$, the measured value would fall within the interval $m \pm 5 \times 10^{-6} m$.

Another critical property of an instrument is its **sensitivity**. Low-sensitivity measurements may fail to detect all anticipated molecules. Conversely, high-sensitivity measurements may lead to a significant amount of **noise** – peaks that do not correspond to any actual molecules.

**Annotation of MS/MS spectra.**   The distribution of masses provided in an MS/MS spectrum contains rich information about the underlying molecular structure. Given an MS/MS spectrum, the aim is to "arrange" the masses into a complete molecular structure. The extent to which MS/MS information is sufficient for deducing the complete structure remains a fundamental open question. However, regardless of the completeness of the structural information, a complex, high-accuracy fragmentation spectrum usually uniquely describes the molecule. The opposite statement is true only under the assumption of a similar MS experimental setup. The same compound can be fragmented in

completely different ways depending on experimental conditions such as ionization adduct species or applied collision energy. This observation rationally motivates the repeated measurement of the same compound with different instrument parameters, thereby enriching the structural information. There are multiple challenges associated with MS/MS annotation and the experimental setup, including the following examples.

First, the width of the isolation window significantly affects the information present in a fragmentation spectrum. A wide window may isolate multiple molecules of similar masses and fragment them, resulting in a single **chimeric spectrum**, which can be misleading for further annotation. Conversely, a narrow window may miss desired isotopes of the same molecule. Ultimately, the isolation window may be triggered for a wrong m/z range, resulting in a spectrum containing nothing but noise.

Second, collision energy affects the number of fragments, their size, and their structure, which in turn affects the number of peaks and their positions. Frequently, as a result of low collision energy, an MS/MS spectrum may contain only a single meaningful peak representing an unfragmented molecule. In contrast, high energy may result in too severe fragmentation, limiting structural annotation.

Finally, CID fragmentation has inherent limitations regarding the interpretation of a molecular structure. For example, it is nearly impossible to distinguish stereoisomers with sole MS/MS data. However, orthogonal sources of information, such as liquid chromatography (LC), may separate the isomers before introducing them to the mass spectrometry stage [38, 39].

## 5 Datasheet

Datasheets for datasets, proposed in [40], aim to serve dataset creators and consumers. They encourage creators to reflect on the creation and maintenance processes, highlighting assumptions, risks, and implications, while providing consumers with essential information to make informed decisions and avoid misuse. Below, we provide a datasheet for the MassSpecGym benchmark.

### 5.1 Motivation

1. **For what purpose was the dataset created?**

   The MassSpecGym benchmark was created to accelerate the process of molecule discovery and identification from tandem mass spectrometry data. To the best of our knowledge, it is the first comprehensive benchmark of its kind.

2. **Who created the dataset (e.g., which team, research group) and on behalf of which entity (e.g., company, institution, organization)?**

   The benchmark was created as a community effort involving multiple research groups in the field of machine learning and computational metabolomics. Please see the list of authors for the complete enumeration. The idea of this benchmark set was conceived at the Dagstuhl Seminar #24181 "Computational 352 Metabolomics: Towards Molecules, Models, and their Meaning".

3. **Who funded the creation of the dataset?**

   The project was funded by national grants listed in the Acknowledgements section. Note that this article solely reflects the opinions and conclusions of its authors and not of its funders.

4. **Any other comments?**

   No other comments.

### 5.2 Composition

1. **What do the instances that comprise the dataset represent (e.g., documents, photos, people, countries)?**

   The instances of the dataset represent tandem mass spectra (measurements of molecules obtained from biological and environmental samples), each annotated with an underlying molecule.

2. **How many instances are there in total (of each type, if appropriate)?**

   The dataset contains 231 thousand instances (tandem mass spectra) annotated with 29 thousand unique molecules in total.

3. **Does the dataset contain all possible instances or is it a sample (not necessarily random) of instances from a larger set?**

   Our MassSpecGym dataset is the largest available of its kind. It is sourced from both publicly available data and our in-house measurements. The dataset provides a diverse sample of theoretically possible tandem mass spectra and molecules, richly covering molecular masses and chemical classes.

4. **What data does each instance consist of?** "Raw" data (e.g., unprocessed text or images) or features? In either case, please provide a description.

   Each instance primarily consists of a tandem mass spectrum and the underlying molecule. Intuitively, a tandem mass spectrum is a set of 2D points (peaks), where each point represents the abundance (intensity) of molecular fragments with specific masses (m/z values). A molecule is represented as a canonical, PubChem-standardized SMILES string. Additionally, each instance contains experimental metadata and auxiliary features derived from mass spectra and molecules. Please see the supplemental materials for the complete list.

5. **Is there a label or target associated with each instance?**

   Yes, molecules define labels to be predicted from tandem mass spectra.

6. **Is any information missing from individual instances?** If so, please provide a description, explaining why this information is missing (e.g., because it was unavailable). This does not include intentionally removed information, but might include, e.g., redacted text.

   Only the instrument type and collision energy metadata features are missing for some of the instances. In these cases, the information was not deposited in the public repositories used as a source for our dataset.

7. **Are relationships between individual instances made explicit (e.g., users' movie ratings, social network links)?**

   No, each instance is considered an independent measurement.

8. **Are there recommended data splits (e.g., training, development/validation, testing)?**

   Our MassSpecGym benchmark provides a data split to standardize the training and evaluation process for different models, making it easily accessible to the broad machine learning community. The data split minimizes the similarity between training and test instances by using molecule edit distance as the measure of similarity.

9. **Are there any errors, sources of noise, or redundancies in the dataset?**

   Noise in signals is inherent to mass spectrometry data. Errors and redundancies are possible in public data sources. However, our standardization pipeline aims to minimize the chances of erroneous and redundant instances.

10. **Is the dataset self-contained, or does it link to or otherwise rely on external resources (e.g., websites, tweets, other datasets)?**

    The dataset is self-contained.

11. **Does the dataset contain data that might be considered confidential (e.g., data that is protected by legal privilege or by doctor–patient confidentiality, data that includes the content of individuals' non-public communications)?** If so, please provide a description.

    No, the dataset does not contain confidential data.

12. **Does the dataset contain data that, if viewed directly, might be offensive, insulting, threatening, or might otherwise cause anxiety?**

    We assume that our mass spectrometry dataset does not contain any information that may cause anxiety.

### 5.3 Collection Process

1. **How was the data associated with each instance acquired?** Was the data directly observable (e.g., raw text, movie ratings), reported by subjects (e.g., survey responses), or indirectly inferred/derived from other data (e.g., part-of-speech tags, model-based guesses for age or language)? If the data was reported by subjects or indirectly inferred/derived from other data, was the data validated/verified? If so, please describe how.

   The data was directly observable based on mass spectrometry measurements.

2. **What mechanisms or procedures were used to collect the data (e.g., hardware appara-tuses or sensors, manual human curation, software programs, software APIs)?** How were these mechanisms or procedures validated?

   The data was collected using mass spectrometry instrumentation, pre-processed by software provided by the instrument vendors, and post-processed using open-source software (mainly matchms). The molecular labels were primarily assigned through manual human curation. These mechanisms were validated by our cleaning and quality assessment workflows.

3. **If the dataset is a sample from a larger set, what was the sampling strategy (e.g., deterministic, probabilistic with specific sampling probabilities)?**

   Our dataset is not a sample from a larger set.

4. **Who was involved in the data collection process (e.g., students, crowdworkers, contrac-tors) and how were they compensated (e.g., how much were crowdworkers paid)?**

   Only the authors of this work were involved in the data collection process.

5. **Over what timeframe was the data collected?** Does this timeframe match the creation timeframe of the data associated with the instances (e.g., recent crawl of old news articles)? If not, please describe the timeframe in which the data associated with the instances was created.

   The spectral libraries used to compile our dataset were downloaded from the official websites on May 27th, 2024.

6. **Were any ethical review processes conducted (e.g., by an institutional review board)?**

   No ethical review was conducted.

### 5.4 Preprocessing/cleaning/labeling

1. **Was any preprocessing/cleaning/labeling of the data done (e.g., discretization or bucket-ing, tokenization, part-of-speech tagging, SIFT feature extraction, removal of instances, processing of missing values)?** If so, please provide a description. If not, you may skip the remaining questions in this section.

   The dataset was preprocessed and cleaned using our standardization pipeline. The pipeline aims to remove noisy and corrupted spectra and ensures reliable molecular labels and metadata. Please see the supplemental materials for details. No manual labeling was performed.

2. **Was the "raw" data saved in addition to the preprocessed/cleaned/labeled data (e.g., to support unanticipated future uses)?** If so, please provide a link or other access point to the "raw" data.

   The "raw" dataset was not saved as part of the MassSpecGym benchmark but can be easily downloaded from the links specified in the supplemental information.

3. **Is the software that was used to preprocess/clean/label the data available?** If so, please provide a link or other access point.

   The preprocessing and cleaning were largely done using the matchms library (`https://github.com/matchms/matchms`). The data processing code is publicly available on our GitHub (`https://github.com/pluskal-lab/MassSpecGym`). The only non-publicly available processing step we performed was the comparison of our mass spectra with the spectra of the commercially available NIST 23 library. This step helped us to partially clean the dataset but NIST 23 spectra were never disclosed or used in any further analyses.

4. **Any other comments?**

   No other comments.

### 5.5 Uses

1. **Has the dataset been used for any tasks already?**

   The dataset has already been used for the three tasks defined for the MassSpecGym bench-mark: *de novo* molecule generation, molecule retrieval, and spectrum simulation.

2. **Is there a repository that links to any or all papers or systems that use the dataset?**

   The links to papers using the MassSpecGym dataset will be available through the Papers with Code resource (`https://paperswithcode.com/`).

3. **What (other) tasks could the dataset be used for?**
   The dataset can be used for other tasks related to the annotation of tandem mass spectra and, more broadly, machine learning from spectra or molecules.

4. **Is there anything about the composition of the dataset or the way it was collected and preprocessed/cleaned/labeled that might impact future uses?** For example, is there anything that a dataset consumer might need to know to avoid uses that could result in unfair treatment of individuals or groups (e.g., stereotyping, quality of service issues) or other risks or harms (e.g., legal risks, financial harms)? If so, please provide a description. Is there anything a dataset consumer could do to mitigate these risks or harms?
   To the best of our knowledge, we do not anticipate any risks or harms resulting from our dataset.

5. **Are there tasks for which the dataset should not be used?** If so, please provide a description.
   We are not aware of any such tasks. However, we anticipate that users will primarily use our dataset for the three mass spectrum annotation tasks defined in the MassSpecGym benchmark.

6. **Any other comments?**
   No other comments.

## 5.6 Distribution

1. **Will the dataset be distributed to third parties outside of the entity (e.g., company, institution, organization) on behalf of which the dataset was created?** If so, please provide a description.
   Yes, the MassSpecGym benchmark (`https://github.com/pluskal-lab/MassSpecGym`) and the MassSpecGym dataset (`https://huggingface.co/datasets/roman-bushuiev/MassSpecGym`) are publicly available.

2. **How will the dataset will be distributed (e.g., tarball on website, API, GitHub)?**
   The training and validation data are available through the Hugging Face Datasets service. The code for training and validation is available via GitHub. An API will be available for evaluation on the test data.

3. **When will the dataset be distributed?**
   This training and validation data are publicly available at the moment of writing.

4. **Will the dataset be distributed under a copyright or other intellectual property (IP) license, and/or under applicable terms of use (ToU)?**
   The data and code are distributed with an MIT license.

5. **Have any third parties imposed IP-based or other restrictions on the data associated with the instances?**
   No third parties have imposed IP-based or other restrictions on the data associated with the instances.

6. **Do any export controls or other regulatory restrictions apply to the dataset or to individual instances?**
   No export controls or other regulatory restrictions apply to the dataset or to individual instances.

7. **Any other comments?**
   No other comments.

## 5.7 Maintenance

1. **Who will be supporting/hosting/maintaining the dataset?**
   The dataset will be supported, hosted and mainted by the authors.

2. **How can the owner/curator/manager of the dataset be contacted (e.g., email address)?**
   The authors can be contacted through the following email addresses: roman.bushuiev@uochb.cas.cz, anton.bushuiev@cvut.cz, josef.sivic@cvut.cz, tomas.pluskal@uochb.cas.cz

3. **Is there an erratum?**

   There is no erratum. If errors are found in the future, they will be released on the project website.

4. **Will the dataset be updated (e.g., to correct labeling errors, add new instances, delete instances)?**

   Yes, if necessary to ensure high quality, the dataset will be updated with new versions on GitHub and Hugging Face Datasets.

5. **If the dataset relates to people, are there applicable limits on the retention of the data associated with the instances (e.g., were the individuals in question told that their data would be retained for a fixed period of time and then deleted)?**

   The dataset does not relate to people.

6. **Will older versions of the dataset continue to be supported/hosted/maintained?** If so, please describe how. If not, please describe how its obsolescence will be communicated to dataset consumers.

   Yes, we plan to maintain the benchmark and update it to new versions, primarily when more publicly available MS/MS data becomes accessible.

7. **If others want to extend/augment/build on/contribute to the dataset, is there a mechanism for them to do so?**

   Yes, users can extend, augment, build on, or contribute to the dataset and the source code by using the standard mechanisms of issues and pull requests on GitHub and Hugging Face Datasets.

8. **Any other comments?**

   No other comments.