# OpenReview forum: "MassSpecGym: A benchmark for the discovery and identification of molecules"
_NeurIPS.cc/2024/Datasets_and_Benchmarks_Track — NeurIPS 2024 Track Datasets and Benchmarks Spotlight_

### Official Review · Reviewer_TGFc · 2024-07-15
**Need improvements**

**Rating:** 7
**Confidence:** 3
**Correctness:** Yes
**Clarity:** Yes

**Review:**

The paper is of high quality with clear methodologies and original contributions, significantly advancing molecular discovery and identification through a comprehensive MS spectrum dataset and innovative evaluation metrics.

Pros:
One significant contribution of this work is the provision of a diverse and highly-standardized tandem MS spectrum dataset, which addresses a critical need in both the ML and MS spectrum.

Provide clear, step-by-step instructions for MS data collection and processing.

The benchmark of three representative MS spectrum-related tasks provides important insights into the applications of this dataset.

Cons:

The cons of this work are listed in the opportunities for improvement.

**Strengths:**

The submission introduces a large-scale, high-quality MS spectrum dataset and evaluation protocols, addressing a critical need in the ML and chemistry communities, with the potential to advance molecular discovery and identification. It provides a comprehensive benchmark for three representative MS/MS spectrum-related tasks, making these tasks accessible to a broader research audience.

**Additional Feedback:**

Null

**Documentation:**

Yes

**Limitations:**

Yes

**Opportunities For Improvement:**

Conduct experiments to compare the performance of tasks using this new dataset against existing datasets to highlight the differences and advantages.

While challenging tasks like molecule generation and retrieval are included, designing new metrics may be necessary to effectively measure performance differences among ML models, such as incorporating largest fragment matching tailored to the problem.

Expand the range of models evaluated for molecule retrieval and diversify the metrics used for spectrum simulation. Additionally, include results from computational simulations for spectrum simulation to provide a comprehensive evaluation.

Broaden the scope of the research to discuss the importance of the selected three representative tasks and their relevance in critical areas such as drug discovery and molecule synthesis. This can demonstrate the broader impact and applicability of the dataset in advancing both scientific research and industrial applications.

**Relation To Prior Work:**

Yes

**Summary And Contributions:**

MassSpecGym introduces the large scale high-quality MS spectra benchmark and standardizes three challenges with new evaluation metrics and a novel data split, facilitating robust molecular discovery and identification. This comprehensive benchmark aims to advance both biomedical and chemical sciences by making MS annotation tasks accessible to the machine learning community.

---

> ### Author Rebuttal · Authors · 2024-08-17
>
> 1\. Need for a new dataset
>
> To the best of our knowledge, the only currently available dataset providing training-validation-testing split is MIST CANOPUS [26] (please see Table 1 in the main paper for details). While this dataset was a valuable first effort in constructing a benchmark for MS/MS spectra annotation, it has notable limitations. A recent reusability report on MIST [27] shows that training on the MIST CANOPUS benchmark leads to overfitting, resulting in significantly lower performance on a held-out dataset. Moreover, the dataset is limited in size, comprising 8K MS/MS spectra and 7K molecules. In contrast, MassSpecGym contains 231K MS/MS spectra and 29K molecules. We also empirically demonstrate the advantages of our dataset. We retrained two different models on two distinct challenges using MIST CANOPUS training data and evaluated them on the test fold of MassSpecGym. To prevent data leakage, we excluded any training examples where the molecules had an MCES distance of less than 10 to any of the test molecules.
>
> Retrieval challenge results:
>
> ||Hitrate@1|Hitrate@5|Hitrate@20|MCES@1|
> |-|-|-|-|-|
> |FingerprintFFN (MIST CANOPUS)|0.016 (0.013-0.020)|0.055 (0.049-0.060)|0.152 (0.143-0.160)|26.76 (26.46-27.06)|
> |FingerprintFFN (MassSpecGym)|**0.025 (0.022-0.030)**|**0.076 (0.070-0.083)**|**0.200 (0.190-0.210)**|**24.66 (24.38-24.96)**|
>
> De novo generation results:
>
> ||Accuracy@1|MCES@1|Tanimoto@1|Accuracy@10|MCES@10|Tanimoto@10|
> |-|-|-|-|-|-|-|
> |SELFIES Transformer (MIST CANOPUS)|0.0|40.21 (39.86-40.56)|0.08 (0.08-0.08)|0.0|27.14 (26.90-27.37)|0.13 (0.12-0.13)|
> |SELFIES Transformer (MassSpecGym)|0.0|**33.28 (32.98-33.58)**|**0.1 (0.10-0.10)**|0.0|**21.84 (21.68-22.02)**|**0.15 (0.15-0.15)**|
>
> 2\. More metrics for generation and retrieval
>
> We thank the reviewer for highlighting the challenging nature of the generation and retrieval tasks and for emphasizing the need for well-suited metrics to effectively measure performance differences among ML models. For both tasks, we have provided robust metrics specifically tailored to the mass spectrometry domain. For example, we defined accuracy in terms of 2D InChI key connectivity blocks to account for 3D isomers that are typically indistinguishable with MS/MS. Similarly, the MCES metric measures the edge edit distance between molecular structures, making it robust to structural rearrangements that might not be distinguishable with MS/MS while also providing additional interpretability. We chose not to include metrics tailored to input mass spectra, such as the suggested largest fragment matching, because molecular fragments associated with individual peaks typically cannot be reliably annotated [28]. We would greatly appreciate it if the reviewer could clarify or correct our understanding of the suggested metric if we have misinterpreted it.
>
> 3\. More models for retrieval and more metrics for simulation
>
> We incorporated the DeepSets baseline (and DeepSets + Fourier features) for molecule retrieval (Table B). DeepSets processes raw 2D peak representations, providing a complementary approach to FingerprintFFN and the state-of-the-art MIST, which operate on binned spectra and predict formulae, respectively.
>
> We added new retrieval results for the simulation models (Table C). For each query spectrum, the simulation model is tasked with predicting a spectrum for each candidate structure. The candidates are then ranked by the cosine similarity of their predicted spectra with the query spectrum. This is a standard approach used in many works [29-31]. We also evaluated baseline performance using additional metrics as alternatives to cosine similarity, which are shown in the common response.
>
> 4\. Importance of the selected three representative tasks
>
> Below, we motivate the choice of three representative tasks and emphasize their importance in various scientific domains.
>
> ***De novo*** **molecule generation.** The first challenge is the *de novo* prediction of a molecular graph from an MS/MS mass spectrum. This challenge can be compared to the goal of AlphaFold, but instead of predicting protein structures from their sequences, the task here is to predict small molecule structures from their MS/MS spectra. As such, this task represents a grand challenge in computational mass spectrometry, given its potential to drive the discovery of novel natural products, drug metabolites, environmental transformation products, and other crucial molecules. A model that can accurately predict molecular structures from MS/MS spectra could significantly advance our understanding of biology and discovery of new drugs by enabling the annotation of metabolomes of uncharacterized organisms.
>
> **Molecule retrieval**. The second challenge focuses on retrieving a molecular graph from a molecular database given a mass spectrum, rather than generating a completely new molecule. This scenario is common in practice when determining if a sample contains specific compounds, such as pesticides, environmental pollutants, or other known substances. This approach is also relevant in drug design, particularly in affinity selection–mass spectrometry, where protein binders are identified from combinatorial libraries of ligands.
>
> **Spectrum simulation.** The third challenge, called spectrum simulation, is the inverse problem of predicting a mass spectrum from a molecular graph. This task has two main motivations. First, it enhances the understanding of MS/MS fragmentation mechanisms in organic chemistry, leading to more precise predictions of how molecules will behave under various conditions. This insight aids in the design of novel compounds and the optimization of synthetic pathways. Second, it allows for generating synthetic spectra by pseudolabeling, thereby expanding the data available for spectral library lookup-based molecule annotation and training machine learning models.

---

> > ### Comment · Reviewer_TGFc · 2024-08-19
> >
> > Thank you for taking the time to incorporate my feedback into the project. I am pleased to see that my concerns have been effectively addressed.

---

### Official Review · Reviewer_ga1S · 2024-07-21
**Badly needed contribution for a very important problem**

**Rating:** 8
**Confidence:** 5

**Review:**

See following sections.

**Strengths:**

* This submission provides an important and badly-needed contribution to the field of computational mass spectrometry - a problem domain that is fundamentally extremely amenable to ML, yet has presented large barriers to entry due to closed-source datasets, poor curation of open data, and considerable domain knowledge requirement. This contribution lowers these barriers to entry considerably and I am very pleased to see an effort like this.
* To date, no open-source MS/MS datasets have been available that match the scale of commercial libraries. This paper solves that issue.
* The approach to preprocessing and curation is rigorous and follows standard practices within the MS community.
* It also clearly formulates three canonical ML problems in tandem mass spectrometry.
* The splitting mechanism advocated is the correct choice for this problem domain.
* The approach to evaluation is strong.

**Additional Feedback:**

I have no additional feedback.

**Clarity:**

The paper is very clear, and written in a manner that should be readily comprehensible outside the mass spectrometry community. A thorough and accessible explanation of MS is provided in the supplement.

**Correctness:**

* Tables 2-4 need error bars.
* Lines 229-230: of the spectra collected, how many did not have collision energy annotations?
* Lines 238-239, re: m/z binning: "These values are precise enough to retain important information about peak accuracy without becoming overly sensitive to measurement." Could the authors share any analysis that motivates this decision?
* Line 240: are precursor peaks annotated as such in the dataset? More broadly, are formula / adduct hypotheses provided along with peaks in this dataset (as they are e.g. in NIST-20)? This would be an important contribution, as a number of recent methods do represent peaks as formulas. I *strongly encourage* the authors to provide the peak annotations from their mass decomposition as part of their dataset.
* Line 262: how many spectra are excluded by this filtering criterion? It would be helpful to understand whether this filtering is disproportionately affecting any chemical classes.
* Supplement 200: how many molecules were excluded by filtering out charged SMILES?

**Documentation:**

The dataset preparation protocol is well documented, and the dataset is provided via HuggingFace.

**Ethics:**

None.

**Limitations:**

* Line 231-232, supplement 205-206: the NIST-20 dataset contains a large number of negative-mode MS/MS spectra. This is a limitation that should be stressed. How many negative-mode spectra were excluded by this filtering decision?
* Different mass spectrometers (even within the two categories of Orbitrap / QTOF) can have different MS2 m/z scan ranges. If this parameter varies, and is not known, it presents a source of noise in this data. Did the authors look into this?
* Similarly, the width of the MS1 isolation window will determine the presence of isotopes in the MS2 spectrum. (This phenomenon is visible in the NIST-20 dataset.) This presents another source of technical noise that was not discussed.
* GC-EI-MS spectra present a growing area of interest for machine learning. The average ML reader may not be aware these are different from LC-ESI-MS/MS spectra; it should be mentioned that this dataset only covers the latter.

**Opportunities For Improvement:**

My critiques relate mainly to modelling MS/MS peaks as formulas, and are detailed below. Overall I would like the authors to provide more information about how many (molecules, spectra) are eliminated at each step in their filtering, which is fairly involved.

**Relation To Prior Work:**

The work is well situated in the context of its field. The authors may consider citing the following publications on lines 49-50:
* Overstreet et al, "QC-GN2oMS2: a Graph Neural Net for High Resolution Mass Spectra Prediction." JCIM 2024
* Zhu and Jonas, "Rapid Approximate Subset-Based Spectra Prediction for Electron Ionization–Mass Spectrometry." ICML 2023
* Murphy et al, "Efficiently predicting high resolution mass spectra with graph neural networks." Anal. Chem. 2023

**Summary And Contributions:**

The paper contributes a curated dataset of tandem mass spectra of small molecules, labelled with molecular structures, and a larger dataset of unlabelled mass spectra; and demonstrate a number of model evaluations on these datasets.

---

> ### Author Rebuttal · Authors · 2024-08-17
>
> 1\. Reduction data at each filtering step
>
> Table S1 in the rebuttal pdf shows the number of spectra removed or modified at each filtering step. The filter names with underscore symbols are part of the matchms cleaning pipeline, while the last four rows correspond to additional postprocessing steps. We will include this table in Supplementary Information.
>
> 2\. Exclusion of negative mode
>
> 153,321 negative-mode spectra were excluded. Since our goal was to create a unified benchmark that does not require specialized mass spectrometry expertise, we decided to include only positive-mode spectra. Incorporating negative-mode spectra would have necessitated, for example, splitting each challenge into two subchallenges or alternatively introducing an additional binary feature (positive or negative mode) in every method, complicating the dataset. We will mention this limitation and opportunity for future work in the discussion.
>
> 3\. M/z scan ranges
>
> We acknowledge that our training data may include variability in m/z scan ranges and other acquisition settings, reflecting the diversity seen in real-world conditions. This diversity is intended to help models trained on our benchmark generalize well to different mass spectra. Unfortunately, detailed m/z scan range metadata is often unavailable, limiting our ability to address this variability more directly. We will emphasize this potential heterogeneity of our dataset in the text.
>
> 4\. MS1 isolation windows
>
> Unfortunately, spectral libraries typically do not provide the width of MS1 isolation windows. Therefore, we were not able to properly address this potential issue. Regarding our in-house data (from Brungs et al.), it is very unlikely that our spectra contain multiple isotopes of a precursor ion or are chimeric spectra. The isolation window was set to 1.2 Da and we excluded spectra not corresponding to the highest MS1 signals within the isolation windows. We will mention this potential source of noise in the text.
>
> 5\. Stress absence of GC-EI-MS
>
> We will address this suggestion by adding the following sentence to the discussion of the limitations of our work: “Our work focuses solely on MS/MS spectra, without considering, for example, electron ionization (EI) or nuclear magnetic resonance (NMR) spectra.”
>
> 6\. Error bars
>
> We enhanced Tables 2-4 with error bars showing 99,9% confidence intervals upon bootstrapping (20,000 resamples). A similar approach was, for example, used in ImageNet [1]and to evaluate AlphaFold2 [2]. Please see Table A-C in the common response.
>
> 7\. How many spectra do not have collision energy annotations?
>
> 109,358 out of 231,104 spectra did not have collision energy annotation. We added the exact numbers to Figure 3b in our paper (named Figure S2 in the rebuttal pdf).
>
> 8\. M/z binning window for simulation metrics
>
> The dataset consists of spectra acquired from multiple kinds of instruments, with varying resolutions. Unfortunately, information about instrument resolution is often unavailable. We employed the precursor-only model as a tool for investigating the effect of binning resolution. Since the precursor-only model only uses the mass of the precursor compound, which we can always compute precisely, measuring the performance of the precursor-only model at different binning widths can provide information about expected performance degradation due to artifacts of discretizing the m/z range at varying levels of granularity.
>
> Judging from the table below, there is a big performance drop when going from bin width of 0.005 Da to 0.002 Da. Based on prior experience, we know that the precursor model should achieve performance around 0.15: it is clear that bin widths <=0.002 Da are not appropriate. We also know that most modern instruments can achieve a resolution of at least 0.01 Da, and ideally we would want to evaluate our models at resolutions that represent practical use cases. Given these arguments, we think 0.01 Da is a reasonable choice, although 0.005 Da could also be appropriate.
>
> | Bin Width (Da) | Precursor-Only Performance (Cosine Similarity) |
> | :---: | :---: |
> | 0.1 | 0.167 |
> | 0.05 | 0.166 |
> | 0.02 | 0.160 |
> | 0.01 | 0.154 |
> | 0.005 | 0.144 |
> | 0.002 | 0.119 |
> | 0.001 | 0.093 |
>
> Additionally, some baseline models we use, such as FFN Fingerprint and GNN, don’t scale well with finer bin resolutions. For instance, reducing the bin size from 0.01 Da to 0.001 Da would cause a 10x increase in the model’s parameters, making the model impractically large given the limited training data. However, models designed to handle such scaling, like FraGNNet(which predicts sparse peak locations), don’t face this issue.
>
> 9\. Annotation of peaks with subformulae
>
> We thank the reviewer for the valuable suggestion to make our benchmark more broadly accessible. We have computed the SIRIUS fragmentation trees for the entire MassSpecGym dataset and uploaded them to our [HuggingFace Hub repository](https://huggingface.co/datasets/roman-bushuiev/MassSpecGym/blob/main/data/auxiliary/MassSpecGym\_fragmentation\_trees.zip). The fragmentation trees annotate spectral peaks with subformulae of a precursor formula.
>
> 10\. How many spectra are excluded by checking peaks not explained with chemical formulae?
>
> This filtering step removed 135,190 spectra, corresponding to only 926 unique structures. The point of this filter was to detect wrong annotations in the public repositories. If most of the fragments cannot be explained by molecular formula decomposition, it is very likely an incorrect annotation, or the spectra contains mostly noise.
>
> 11\. How many molecules were excluded by filtering out charged SMILES?
>
> 4776 spectra were excluded by filtering out charged SMILES.
>
> 12\. Additional references
>
> We appreciate the reviewer’s suggestion of relevant references and will cite them on lines 49-50.

---

> > ### Comment · Reviewer_ga1S · 2024-08-24
> >
> > Thank you for thoroughly addressing my feedback. I am satisfied with the rebuttal and have no further comments.

---

### Official Review · Reviewer_FppL · 2024-07-24

**Rating:** 6
**Confidence:** 4
**Correctness:** Yes
**Clarity:** Yes

**Review:**

Pros:
•	The paper introduced a realistic research problem for AI community, that is to deduce molecules from the MASS spectrum.
•	The dataset collected is comprehensive.

Cons:
•	The proposed tasks are not well-formulated. In de novo molecule structure design, why directly generating molecule graph from the 2D MASS spectrum is infeasible? Any empirical experiment results or case study? what is the reason to provide chemical formula? Is this information generally available for the task?
•	Missing baselines. For the de novo molecule structure design task, one SMILES transformers is tested, however, there are a bunch of graph generation baselines, the author are encouraged to test the effectiveness of these models in the benchmark
•	Standard deviation was not reported.
•	The future work is not well discussed. What is the gap for current AI model struggling in these tasks? And any potential solution to fill the gap?

**Strengths:**

The paper presents an interesting benchmark in understanding the relationship between MASS spectra and molecule structure. This benchmark opens a new problem for AI models to solve chemical tasks. The public available data will be beneficial to the community of both AI and chemistry.

**Additional Feedback:**

None

**Documentation:**

Yes

**Limitations:**

The main limitation is the lack of significance of contribution with the current work. More in-depth and insightful discussion could strengthen the value of this work.

**Opportunities For Improvement:**

•	It would be better to include more baselines.
•	It would be better to see other representations, for example, generating molecules as a SMILES or SELFIES string.

**Relation To Prior Work:**

Yes

**Summary And Contributions:**

This paper aims at benchmarking AI models in solving molecule structures from the MASS spectra (a spectrum that encodes important information for a molecule). The key contributions of this work include the following:
•	A comprehensive MASS spectra dataset (231,000 MASS spectra from 29,000 molecules).
•	Definition of three new tasks for machine learning models: de novo molecular structure generation, molecule retrieval, and spectrum simulation.
•	Application of several machine learning algorithms, revealing that  current models struggle in all the proposed tasks.

---

> ### Author Rebuttal · Authors · 2024-08-17
>
> 1\. Directly generating molecule graph
>
> There have been attempts in the field at *de novo* generation of 2D molecular structures [17], however the reliability of such methods is thus far not yet high enough for most practical applications. Based on the analytical capabilities of mass spectrometry, determining the complete 3D configuration of a molecule is not possible from mass spectra alone. Instead, orthogonal analytical techniques, such as nuclear magnetic resonance (NMR), X-ray crystallography, or those based on infrared spectroscopy, are required to elucidate the full stereoisomerism of a molecule [15]. For example, Figure 1 in [16] shows a mass spectrum that could be equally matched to one of the 14 stereo- and regioisomer molecular structures. The complexity of de novo molecule generation is our motivation for including task two focused on molecule retrieval.
>
> 2\. Providing chemical formula
>
> Indeed, in practice, chemical formulae can be derived with high accuracy by utilizing MS1 mass spectra, an orthogonal data source to MS/MS [18, 19]. Since working with MS1 data is typically based on combinatorial optimization rather than machine learning [20], our benchmark directly provides chemical formulae instead of MS1 spectra, imitating the output of the MS1 spectra processing pipelines. However, we present this scenario as a bonus challenge because chemical formula prediction remains a partially unsolved problem. For example, elements such as fluorine, which have only one stable isotope, cannot be derived from MS1 data alone and still pose challenges even with MS2 data [21].
>
> 3\. Graph generation baselines
>
> Performing *de novo* molecule generation from a mass spectrum by generating its molecule graph rather than using a discrete token-based approach, such as the SMILES transformer baseline that is included, is a highly relevant suggestion. To our knowledge, however, such an approach has not been described in the scientific literature yet, with state-of-the-art solutions using language-based models [17, 22]. While in other BioML research areas many 2D and 3D graph generation models have been developed [23], none are directly applicable to mass spectrometry. The majority of these models focus on unconditional molecule generation or are conditioned on other features, such as protein pockets or molecular motifs. The generation conditioned on a mass spectrum remains an open research problem. In fact, we anticipate that the presented benchmark and challenge will promote the development of exactly such creative solutions to this problem. We are happy to incorporate or compare with any concrete suggested reference.
>
> 4\. Standard deviation
>
> We thank the reviewer for highlighting this point and provide the tables with confidence intervals in the general rebuttal response (Table A, Table B, and Table C). We have chosen to add the 99.9% confidence intervals upon bootstraping (20,000 resamples) to the result tables for all three challenges  rather than standard deviation because confidence intervals are more robust for metrics such as hit rate@k, which takes a value of 0 or 1 for each data point. Such an approach has also been applied in ImageNet [24] and to evaluate AlphaFold2 [25]. We do not provide the table with standard deviations here due to the space constraints. However, we can provide them during the discussion period, if needed.
>
> 5\. Future work
>
> Our future work has two main directions. First, we plan to continuously update MassSpecGym with new public and in-house MS/MS data, including simulated spectra and additional types such as negative ionization mode spectra, multi-stage fragmentation trees (MSn), and electron ionization (EI) spectra. We also aim to expand the challenges to include tasks such as molecular networking, which aims to cluster spectra of structurally related molecules. Second, by progressively enhancing MassSpecGym with advanced methods, we intend to make it a hub for state-of-the-art models in MS/MS spectra annotation. This will empower machine learning researchers to rapidly develop innovative models, a particularly crucial focus given the historically limited collaboration between mass spectrometry experts and AI specialists. As a result, well-established machine learning techniques such as generating molecular graphs via diffusion models or applying domain adaptation across different mass spectrometry systems remain largely unexplored. We anticipate MassSpecGym will enable filling this gap by making benchmarked problems approachable to the machine learning community. Additionally, by offering a user-friendly interface to run these models, we aim to make them readily accessible to life scientists. Therefore, we believe MassSpecGym will play a pivotal role in advancing next-generation machine learning methods, driving significant progress in biomedical and chemical sciences.
>
> 6\. More baselines
>
> We have added several additional baselines. Please see Tables A and B in the common response to the reviewers.
>
> 7\. Other representations
>
> We expand both input and output representations by implementing Fourier features for mass spectra and SELFIES for molecules. Table A shows the performance of DeepSets baseline with and without Fourier features [21]. Table B compares transformer implementations with tokenization based on SMILES and SELFIES.
>
> 8\. The main limitation is the lack of significance of contribution with the current work. More in-depth and insightful discussion could strengthen the value of this work.
>
> In response to this valuable feedback, we have significantly revised the main text to place our contributions within a broader context, emphasizing their significance (please refer to the common rebuttal response). The key value of our work lies in making problems of discovery and identification of molecules (such as new bioactive molecules/drugs or environmental pollutants) easily accessible to machine learning researchers who may lack expertise in mass spectrometry.

---

> > ### Comment · Reviewer_FppL · 2024-08-27
> >
> > Thank you for the response, which addressed my questions.

---

### Official Review · Reviewer_wDbm · 2024-07-24
**Facilitating discovery and identification of molecules**

**Rating:** 7
**Confidence:** 3
**Clarity:** The paper is well written and underst…

**Review:**

This study addresses a prominent issue in metabolomics, namely the lack of standardized data and community benchmarks for evaluating algorithms. The dataset comprises widely used existing databases combined with in-house data. However, individual algorithms standardize the data for benchmarking using their metrics. The study introduces a standardized benchmarking approach based on these metrics. With its emphasis on dataset size and standardization, the paper presents a novel contribution to the advancement of MS-based metabolomics.
The given data cleaning steps are fair for cleaning spectra for further analysis. The paper explains clearly the majority of the paper. However, there are some points that I have questions.
- How many spectra have all metadata available? It would be great to determine the number of spectra used for each data splitting method outlined in Supplementary Material Table 2.
- In the supplementary material, for the data splitting, the composition is given as 55%-22%-22% when addressing the issue of the underrepresentation of molecules in the training set. Does this composition correspond to the "Clusters" approach in Table 2? If it does, why does reassigning samples from validation and test sets to the training set reduce the percentage of training?
- In the supplementary material, the data splitting seems to be based on clusters to avoid sampling bias and/or data leakage. Is it acceptable to use a 41% training set in this case? Additionally, please assess the potential for data leakage in the spectra and molecular splitting as presented in Table 2? Also, when mentioning that the "composition in terms of molecular structure" for the splitting in line 303, was the "2D InChI key" of molecules used for splitting?
- The authors show that the combined datasets offer extensive coverage. I'm curious about the distribution of molecule classes within the splits. A bar plot showing the distribution would be helpful in understanding if certain molecule classes have a higher representation in the splits than others. Further, maybe there is an opportunity to create a split that would results in a leave-on-out type approach for certain classes to assess generalization beyond the classes a model was trained on?
- In the Supplementary Material, specifically in the section on collision energy standardization, there's a reference to line 267, which states, "Spectra with ramped collision energies typically report the ramp range (i.e. 10-30 eV): for simplicity, we represent this data using the simple average of these boundary points (i.e. 20 eV)." I'm uncertain about the validity of simplifying the representation in this manner and its potential impact. Do the authors have data supporting this approach, e.g. compare the ramped spectrum against a spectrum that was acquired at the mean? Further, collision energy was reported to not be stable across machines and even on the same machine over time. That would suggest that further metadata is required to ensure that collision energy “means” the same over the collected dataset.
- Can the authors provide a suggestion for which data splitting is recommended for which task? How does the split effect the results of the benchmarks? Further, more insight into the performance of methods may be gained by splitting the performance measure for mass spectrometer, collision energy, molecule class. On this note, a more detailed description of the distribution and biases introduced by this would be helpful to assess where the resource has gaps and where the ingestion of additional data would be beneficial.
- The authors attempted to benchmark existing tools. However, the number of tools tested is small and largely contains random predictors. While the authors state that commercial software cannot be used, I wonder if this list could not be extended. CASMI and some google search shows a number of available tools (i.e. web services). While it may not be 100% representative (because data leakage cannot be ruled out), could the authors use the holdout data from Brungs et al. and evaluate existing tools, without retraining on a new split? This would expand the benchmarking section and would give an indication of performance and what there is to gain with better/more advanced models.

**Strengths:**

The study strength is ‘standardized dataset and evaluation process for different models’. It makes a broader range for machine learning and computational metabolomics. The tasks described cover a nice range of complexity as well as use cases, thus, the dataset has the potential for becoming the new standard for developing tools and models.

**Additional Feedback:**

No additional feedback.

**Correctness:**

The claims made in the submission are supported by data and the dataset appears to be constructed in a sound way. The evaluations are meaningful.

**Documentation:**

The collection, processing and dataset is well described. However, I believe the authors have an opportunity here to extend the documentation of the GitHub repo, which – as far as I could tell – only contains the documentation of the landing page. No notebooks I checked contained any documentation and neither is a documentation of the source code available. This may hinder adoption and extension.

**Ethics:**

No ethics concerns.

**Limitations:**

The authors did not mention any limitations in the conclusion. Some may have been brought up here, but I strongly encourage the authors to extend this section. For example, the class imbalance (e.g. Fig 3c).

**Opportunities For Improvement:**

Please see the opportunities for improvement under Review.

**Relation To Prior Work:**

Previous work is highlighted and put in context of this work.

**Summary And Contributions:**

In their study, Bushuiev et al. tackle the issue of the lack of standardized pre-processing or data handling protocol in MS-based metabolomics, which has hindered the development of dedicated methods. To address this, they introduce three tasks for benchmarking and a publicly available standardized MS/MS dataset. They curated a dataset by combining existing databases (MONA, GNPS, and MassBank) and in-house data from Brungs et al., yielding 231k high-quality mass spectra representing 29k unique molecular structures. The team conducted rigorous pre-processing steps, employing spectrum-based and metadata-based filtering, as well as quality assessment for data cleaning. Additionally, they standardized the data by applying spectrum, instrument, collision energy, and molecular structure standardization. Bushuiev et al. also present a training-validation-test split metric utilizing the maximum common edge subgraph-based clustering technique. Their study includes benchmarking for current challenges in three MS/MS annotation tasks: de novo molecular structure generation, molecule retrieval, and spectrum simulation. In each of these areas, they introduce novel metrics for evaluating new algorithms.

---

> ### Author Rebuttal · Authors · 2024-08-17
>
> 1\. Availability of metadata and data splits
>
> All spectra included in the MassSpecGym dataset have adduct, precursor m/z and molecular labels. Only collision energy and instrument type is missing for some of the spectra. 53% of spectra have normalized collision energy available and 98% of spectra have instrument type information available. We added the exact numbers to the updated version of Figure 3b from the original paper (Figure S2 in the common rebuttal pdf).
>
> Please note that Supplementary Table 2 describes the fold composition of a single split obtained using MCES-based hierarchical clustering, which corresponds to our final MassSpecGym benchmark (providing a single strictly defined split). To show the exact number of spectra, we have added the numbers corresponding to the percentages.
>
> ||**Num.spectra**|**Num.molecules**|**Num.clusters**|
> |-|-|-|-|
> |**Training**|194,119(84%)|25,046(79%)|3,061(41%)|
> |**Validation**|19,429(8%)|3,386(11%)|2,221(30%)|
> |**Testing**|17,556(8%)|3,170(10%)|2,202(29%)|
> |All metadata available|||||
> |**Training**|101,573(84%)|13,543(74%)|2,628(41%)|
> |**Validation**|9,975(8%)|2,445(13%)|1,917(30%)|
> |**Testing**|10,159(8%)|2,417(13%)|1,907(30%)|
>
> 2\. Clarification of 55%-22%-22% split composition
>
> The composition corresponds to “Molecules” in the original Table 2 (55%-22%-22% were reassigned to 74%-13%-13%). We will accordingly clarify the analysis of our split in the text.
>
> 3\. Is it acceptable to use a 41% training set in this case?
>
> We thank the reviewer for highlighting this important point which was not well discussed in our text. We would like to clarify that we did not use 41% of spectra for the training set, but used 41% of the selected clusters, which corresponds to selecting 84% of the spectra. Due to our strict clustering cutoff of MCES \< 10, the majority of molecules were assigned to a single, largest cluster comprising 18,483 molecules. While these molecules are structurally distinct, they are transitively connected by an MCES \< 10 into a single cluster. Since this largest and diverse cluster is part of the training fold, aiming for a good split of the number of spectra and molecules resulted in only having 41% of the clusters in the training set.
>
> 4\. Leakage under different splits
>
> The data leakage resulting from a 2D InChI key-based split is illustrated in Figure 2 of the main text. The figure does not show spectrum-based splitting because it is very similar to a random split with respect to molecules, as 84% of the molecules in MassSpecGym have more than one spectrum.
>
> 4\. Molecular classes: distribution in folds and splitting
>
> In the rebuttal pdf, we provide a bar plot (Figure S1) showing the balanced distribution of chemical classes across folds. We considered various data-splitting options, such as by mass spectrometry conditions or molecular properties, including chemical classes. However, since our primary audience is machine learning researchers who may not be familiar with mass spectrometry or molecular structure details, we chose a single, clearly defined split. While MCES distance is easily understood as a graph edge edit distance, chemical classes might not be as intuitive for the computer science community.
>
> 5\. Ramped collision energies
>
> As we progressively refined our cleaning pipeline for mass spectra, this part of the text became irrelevant because all spectra with ramped collision energies were filtered out. We apologize for the confusion and will remove the text.
>
> 6\. Splitting performance by metadata
>
> Figures S3-5 in the rebuttal pdf provide the performance of the retrieval methods grouped by instrument types, collision energies, and chemical classes. We will include similar results for other challenges and metrics in Supplementary Information.
>
> 7\. Benchmarking of commercial tools
>
> While we appreciate the suggestion to expand the benchmarking by including commercial tools, our primary goal was to establish an open-source benchmark that strictly eliminates data leakage. Introducing tools trained on different proprietary datasets could compromise this objective. We believe that the reported performance could be potentially misleading, especially to newcomers to the field of computational metabolomics. Instead, we are focusing on reimplementing SIRIUS, a state-of-the-art commercial software, and retraining it on MassSpecGym data, which will provide a fair and accurate evaluation. Please note that we have included more baselines into evaluation (Table A and Table B in the common response to all reviewers).
>
> 8\. Limitations
>
> MassSpecGym aims to make machine learning applied to MS/MS spectra accessible to the machine learning community and rigorously standardized. To achieve this, we had to make certain simplifications that inherently limit MassSpecGym. For example, we focused exclusively on MS/MS spectra acquired in positive ionization mode, retained only spectra with the most common ionization adducts and with singly-charged ions, and sought to eliminate noisy spectra. However, this approach does not fully reflect real-world scenarios where a significant portion of spectra may be instrument noise or acquired in different settings. Additionally, we were unable to adequately address certain types of noise, such as isotope signals in MS/MS spectra or chimeric spectra representing multiple molecules, due to the lack of appropriate metadata in public spectral libraries. Our dataset is also combined from highly heterogeneous and imbalanced spectral libraries, which may pose challenges when training machine learning models. Finally, our work is solely focused on MS/MS spectra, not considering other types of spectra, such as electron ionization (EI) or nuclear magnetic resonance (NMR) spectra.
>
> 9\. Documentation
>
> We are in the process of creating a “Read the Docs” documentation page to improve the accessibility and improved documentation of the code. We will continue to enhance the documentation during and after the rebuttal.

---

### Author Rebuttal · Authors · 2024-08-17

We thank the reviewers for their constructive and insightful feedback. Below we summarize the main strengths of our paper as they have been pointed out by the reviewers as well as how we address the main weaknesses.

**Strengths**

* The paper presents a practical AI research problem with significant community impact (FppL, ga1S), addressing a critical need in the field (ga1S, TGFc).
* The paper is well-written and clear (wDbm, ga1S, TGFc).
* The dataset construction methodology is clear and rigorous (wDbm, ga1S, TGFc).
* The constructed dataset is comprehensive (FppL, TGFc).
* The data splitting mechanism is the correct choice (ga1S).
* The tasks cover a nice range of complexity and use cases (wDbm), are canonical in the field (ga1S) and provide important insights into the applications of the dataset (TGFc).
* The evaluation metrics and innovative (TGFc).

**Weaknesses**

We address all the points mentioned by the reviewers in our individual responses below. Here, we would like to highlight the following two:

* Broader context

We substantially revised the paper to emphasize the significance of the benchmark. First, we broadened the scope of research discussed in the first paragraph, updated below. Second, we provided a stronger rationale for the choice of challenges, as detailed in the response TGFc 4. Third, we included discussions on limitations and future work in FppL 5 and wDbm 8.

> The discovery and identification of small molecules profoundly influence numerous scientific fields, including organic chemistry [1], molecular biology [2], drug development [3], disease diagnosis [4], environmental analysis [5], and space exploration [6]. Despite significant progress, it is estimated that only a small fraction of molecules across the kingdoms of life have been discovered [7]. Tandem mass spectrometry (MS/MS) is the primary technique for elucidating molecular structures from biological and environmental samples, supporting a wide range of applications in biotechnology and medicine [8]. In drug development, MS/MS is crucial for identifying novel bioactive compounds [9], such as those targeting cancer and infectious diseases [10]. MS/MS also plays a key role in clinical settings for determining appropriate drug dosages and assessing potential side effects [11]. In environmental analysis, it enables the detection of pollutants, which is vital for monitoring environmental health [12]. Moreover, MS/MS addresses various challenges in structural biology, including the discovery of ligands that bind to target proteins [13] and the elucidation of metabolic pathways [14].

* Baselines and error bars

We incorporated additional baselines and data representations and are in the process of including more. We added 99.9% confidence intervals as error bars. We conducted two additional baseline evaluation experiments suggested by the reviewers (TGFc 1, wDbm 6).

**Table A. Results for the *de novo* molecule generation challenge.**
||Accuracy@1|MCES@1|Tanimoto@1|Accuracy@10|MCES@10|Tanimoto@10|
|-|-|-|-|-|-|-|
|Random chemical generation|0.0|**28.59 (28.33-28.84)**|0.07 (0.07-0.07)|0.0|25.72 (25.49-25.96)|0.1 (0.10-0.10)|
|SMILES Transformer|0.0|53.8 (52.95-54.65)|0.07 (0.07-0.08)|0.0|21.97 (21.78-22.16)|**0.17 (0.17-0.17)**|
|SELFIES Transformer|0.0|33.28 (32.98-33.58)|**0.1 (0.10-0.10)**|0.0|**21.84 (21.68-22.02)**|0.15 (0.15-0.15)|
|Bonus chemical formulae challenge|||||||
|SMILES Transformer|0.0|59.65 (59.01-60.10)|0.08 (0.07-0.08)|0.0|18.85 (18.25-19.21)|**0.20 (0.19-0.20)**|
|Random chemical generation|0.0|**21.11 (20.97-21.26)**|0.08 (0.08-0.08)|0.0|18.25 (18.14-18.35)|0.11 (0.11-0.11)|
|SELFIES Transformer|0.0|23.67 (23.14-24.11)|**0.13 (0.13-0.13)**|0.0|**16.99 (16.47-17.60)**|0.19 (0.19-0.19)|

**Table B. Results for the molecule retrieval challenge.**
||Hitrate@1|Hitrate@5|Hitrate@20|MCES@1|
|-|-|-|-|-|
|Random|0.000 (0.000-0.001)|0.020 (0.017-0.024)|0.082 (0.076-0.089)|30.81 (30.43-31.24)|
|DeepSets|0.015 (0.012-0.018)|0.062 (0.056-0.068)|0.192 (0.183-0.202)|25.11 (24.84-25.39)|
|FingerprintFFN|0.025 (0.022-0.030)|0.076 (0.070-0.083)|0.200 (0.190-0.210)|24.66 (24.38-24.96)|
|DeepSets + Fourier features|0.052 (0.047-0.058)|0.126 (0.118-0.134)|0.282 (0.271-0.293)|22.13 (21.85-22.42)|
|MIST|**0.146 (0.138-0.155)**|**0.349 (0.337-0.361)**|**0.592 (0.580-0.603)**|**15.37 (15.12-15.61)**|
|Bonus chemical formulae challenge|||||
|Random|0.031 (0.027-0.035)|0.114 (0.106-0.122)|0.277 (0.266-0.288)|13.87 (13.70-14.03)|
|DeepSets|0.044 (0.039-0.050)|0.145 (0.136-0.154)|0.308 (0.296-0.319)|15.04 (14.89-15.19)|
|FingerprintFFN|0.051 (0.046-0.057)|0.147 (0.138-0.156)|0.320 (0.308-0.331)|14.94 (14.78-15.09)|
|DeepSets + Fourier features|0.066 (0.060-0.072)|0.165 (0.156-0.174)|0.335 (0.323-0.346)|14.14 (13.99-14.30)|
|MIST|**0.096 (0.089-0.103)**|**0.221 (0.211-0.231)**|**0.411 (0.399-0.423)**|**12.75 (12.59-12.91)**|

**Table C. Results for the spectrum simulation challenge.**
||Cosine Similarity|SQRT Cosine Similarity|Jensen-Shannon Similarity|Hitrate@1|Hitrate@5|Hitrate@20|
|-|-|-|-|-|-|-|
|Precursor m/z|0.154 (0.144-0.165)|0.165 (0.156-0.174)|0.409 (0.402-0.415)|0.004 (0.002-0.006)|0.017 (0.013-0.022)|0.072 (0.063-0.080)|
|Fingerprint FFN|0.249 (0.239-0.261)|0.273 (0.264-0.282)|0.478 (0.472-0.484)|0.084 (0.076-0.094)|0.214 (0.200-0.228)|0.386 (0.370-0.402)|
|GNN|0.192 (0.182-0.202)|0.218 (0.210-0.227)|0.442 (0.436-0.447)|0.039 (0.033-0.046)|0.119 (0.109-0.130)|0.263 (0.248-0.278)|
|FraGNNet|**0.519 (0.509-0.531)**|**0.508 (0.501-0.516)**|**0.633 (0.627-0.638)**|**0.466 (0.450-0.482)**|**0.726 (0.710-0.740)**|**0.836 (0.823-0.848)**|
|Bonus chemical formulae challenge|||||||
|Precursor m/z|-|-|-|0.021 (0.017-0.026)|0.085 (0.076-0.095)|0.227 (0.213-0.241)|
|Fingerprint FFN|-|-|-|0.076 (0.068-0.085)|0.227 (0.214-0.241)|0.441 (0.425-0.458)|
|GNN|-|-|-|0.036 (0.030-0.043)|0.136 (0.125-0.147)|0.338 (0.322-0.353)|
|FraGNNet|-|-|-|**0.319 (0.304-0.335)**|**0.632 (0.616-0.647)**|**0.827 (0.814-0.839)**|

---

### Decision · Program_Chairs · 2024-09-26

**Decision:**

Accept (Spotlight)

**Comment:**

In this manuscript, the authors describe MassSpecGym, a new dataset and a benchmark for MS/MS data. The dataset is the largest one to date freely available to the public. The benchmark contains 3 annotation challenges and a number of methods were compared.

Strengths:
1. The reviewers generally found the dataset to be very useful and such a dataset is currently missing in the field.
2. Some reviewers praised the careful data processing, metadata-based filtering, quality assessment, and data cleaning.
3. Most reviewers found the tasks well defined and that they provided useful insights.

Weaknesses:
1. A reviewer commented that the number of methods tested was small -- the authors explained why they chose not to include commercial methods but the issue of having only a small number of methods was not directly addressed.
2. There were a number of questions about splits.
3. There were various clarification questions asked by the different reviewers.

Overall, three reviewers confirmed that the authors had satisfactorily addressed their comments in the rebuttals. One reviewer did not participate in post-review discussions, but based on the authors' rebuttal, most questions appear answered. All four reviewers found the manuscript to be above the acceptance threshold, with their scores ranging from 6 to 8.